# MixtureVitae: Open Web-Scale Pretraining Dataset With High Quality Instruction and Reasoning Data Built from Permissive Text Sources

## Abstract

We present MixtureVitae, an open-access pretraining corpus built to minimize legal risk while providing strong downstream performance. MixtureVitae follows a permissive-first, risk-mitigated sourcing strategy that combines public-domain and permissively licensed text (e.g., CC-BY/Apache) with carefully justified low-risk additions (e.g., government works and EU TDM-eligible sources). MixtureVitae adopts a simple, single-stage pretraining recipe that integrates a large proportion of permissive synthetic instruction and reasoning data—signals typically introduced during post-training and generally scarce in permissive web corpora. We categorize all sources into a three-tier scheme that reflects varying risk levels and provide shard-level provenance metadata to enable risk-aware usage. In controlled experiments using the open-sci-ref training protocol (fixed architectures and hyperparameters; 50B and 300B token budgets across 130M–1.7B parameters), models trained on MixtureVitae consistently outperform other permissive datasets across a suite of standard benchmarks, and at the 1.7B-parameters/300B-tokens setting, they surpass FineWeb-Edu and approach DCLM late in training. Performance is particularly strong on MMLU and on math and code benchmarks: a 1.7B model pretrained on 300B MixtureVitae tokens matches or exceeds a strong 1.7B instruction-tuned baseline on GSM8K, HumanEval, and MBPP, despite using over $36\times$ fewer tokens (300B vs. $\approx$11T). Supported by a thorough decontamination analysis, these results show that permissive-first data with high instruction and reasoning density, tiered by licensing and provenance-related risk, can provide a practical and risk-mitigated foundation for training capable LLMs, reducing reliance on broad web scrapes without sacrificing competitiveness.

## 1 Introduction

The proliferation of large language models (LLMs) has transformed the landscape of artificial intelligence, yet their development often relies on a legally and ethically precarious foundation. The vast majority of performant models are pretrained on massive web scrapes, indiscriminately mixing public-domain content with copyrighted materials such as books, news articles, and personal websites without explicit permission (Raffel et al., 2020; Gao et al., 2020). This practice has led to a growing number of copyright infringement lawsuits, creating significant legal uncertainty for both academic researchers and commercial developers and threatening the future of the field. At the same time, practitioners who wish to avoid this risk have few alternatives, as most high-performing pretraining mixtures rely, at least in part, on opaque or non-permissive web scrapes.

Compounding this uncertainty is the prevailing assumption that state-of-the-art performance is inextricably linked to the sheer scale and diversity offered by these legally ambiguous web scrapes. The absence of a high-performance, large-scale pretraining dataset that actively mitigates these risks has forced a difficult choice between performance and compliance. In practice, the strongest open baselines such as FineWeb-Edu (Lozhkov et al., 2024a) and DCLM (Data-Centric Language Models, 2024) still rely on mixed-license or unspecified web data, whereas strictly permissive corpora tend

to lag behind them on reasoning-heavy benchmarks. This raises a critical question: Can a powerful language model be trained on a dataset that provides a more legally robust foundation?

To this question, we answer "yes": We introduce **MixtureVitae**, a **422**-billion-token, open-access pretraining dataset constructed to minimize copyright risk while explicitly demonstrating that a reasoning- and instruction-dense, permissive-first mixture can substantially close the performance gap to leading non-permissive corpora. The core of **MixtureVitae**'s "permissive-first" data comprise (1) text with clear and permissive licenses (e.g., CC-BY-*, Apache 2.0), public-domain text, and copyright-exempt text such as US federal works (see Appendix G) and (2) risk-mitigated text. Following Phi-4 (Abdin et al., 2024), which shows that the addition of synthetic and web-rewrite data boosts performance, we address the scarcity of organic reasoning and conversational dialogue in strictly permissive sources significantly augmenting **MixtureVitae** with targeted synthetic data, which is derived from permissive models and sources. We call this combination of expressly licensed and risk-mitigated methods the **"permissive-first"** approach.

To validate our approach, we train models with **130M, 400M, 1.3B, and 1.7B parameters** on **MixtureVitae** and compare their performance against several prominent open datasets. The results first confirm that **MixtureVitae significantly outperforms all other permissively licensed baselines**, with the performance gap widening as the model scale increases. The more critical test, however, is against popular non-permissive datasets containing higher proportions of copyrighted or ambiguously-licensed material. In this setting, our models achieve competitive performance, and on math and code benchmarks, our 1.7B base model matches or exceeds a strong 1.7B instruction-tuned baseline despite being trained on a dramatically smaller token budget.

In summary, our contributions are threefold:

- **Permissive-first, risk-mitigated, and performant recipe for pretraining corpora.** We present **MixtureVitae**, the first highly-performant, permissive-first, and risk-mitigated pretraining corpus that deliberately front-loads high-quality reasoning and instruction data to drive capability gains in small models. It is organized into auditable provenance tiers and constructed via a positive-inclusion pipeline, avoiding the need for retroactive filtering.

- **We demonstrate that reliance on indiscriminately scraped, high-risk copyrighted data is not a prerequisite for training capable LLMs.** Leveraging the `open-sci-ref` (Nezhurina et al., 2025) protocol to ensure rigorous comparison across 130M–1.7B parameter scales, we demonstrate the value of front-loading instruction and reasoning data into pre-training. Our 422B-token, permissive-first mixture closes the gap to mixed-license baselines while providing an auditable legal provenance. Furthermore, we show that our 1.7B base model, despite a limited 300B token budget, is comparable across multiple reasoning benchmarks to a strong 1.7B instruction-tuned baseline—trained on roughly $36\times$ more tokens ($\approx$11T).

- **Evaluation integrity and reusable artifacts**. We perform a large-scale 13-gram decontamination analysis across all benchmarks, showing that **MixtureVitae**'s gains persist on decontaminated test sets and when removing shards responsible for most detected overlap, and we release the corpus, shard-level provenance metadata, and curation code to enable compliant, reproducible pretraining in future work.

## 2 DATASET

We adopt a permissive-first, risk-mitigated strategy, combining sources with clear permissive licenses (e.g. CC-BY, Apache, public domain) with narrowly justified inclusions (government works, EU TDM-eligible data) and targeted synthetic data. Within this framework, the **MixtureVitae** dataset is constructed from three primary categories: curated sources for domain-specific expertise, diverse web data for language and general knowledge and instruction-following and reasoning datasets to enhance reasoning and task-completion abilities.

The major categories of our corpus are visualized in Figure 1a. We provide a granular breakdown showing the token count for each component (Figure 4), the license distribution (Figure 1b), and synthetic data usage (Figure 2a). Specific data sources are detailed in the following subsections.

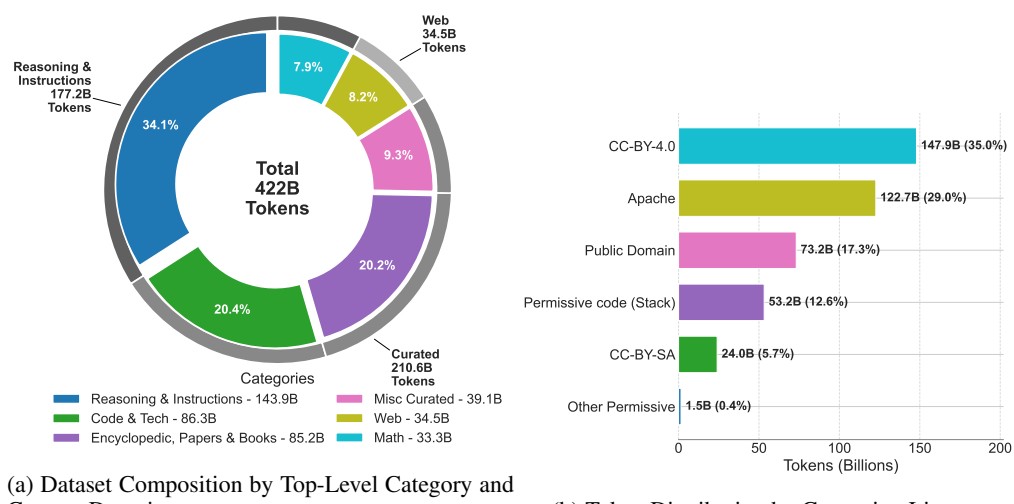

(a) Dataset Composition by Top-Level Category and Content Domain

(b) Token Distribution by Governing License

Figure 1: Composition of the **MixtureVitae** dataset (permissive-first, risk-mitigated composition).

## 2.1 DATA SOURCES

Our dataset selection process is governed by a two-layer criteria, prioritizing risk mitigation followed by quality and capability objectives:

- **Legal & Licensing:** The primary filter is legal compliance. A dataset is considered only if it operates under a clear permissive license (e.g., CC-BY, Apache 2.0) or is in the public domain. For synthetic data, we further scrutinize the provenance of seed corpora and generator models (Appendix J). The majority of our synthetic sources satisfy full provenance transparency (classified as Tier 1), while a minority of community reasoning datasets with opaque provenance are categorized as Tier 2 to manage residual risk.

- **Quality & Capability:** Among compliant sources, we prioritize datasets with prior evidence of high performance in community mixtures (e.g., Soldaini & Lo, 2023). Furthermore, to address the reasoning deficits typical of strictly permissive web scrapes, we target high-density instruction and reasoning data, a choice driven by the need to boost performance on tasks such as GSM8K (Cobbe et al., 2021a) and MMLU (Hendrycks et al., 2021).

The following sections describe each of the three categories of data in **MixtureVitae**: web, curated sources, and instruction and reasoning datasets.

### 2.1.1 WEB-SCALE CORPORA

One subset of our pre-training data is derived from web-scale datasets including Nemotron-CC (Cicero et al., 2024), MagaCorpus (Xu et al., 2024b), and FineFineWeb (Zhang et al., 2024). It also contains synthetic data generated by rephrasing web text from Nemotron-CC and MagaCorpus.

### 2.1.2 CURATED DATASETS

To incorporate domain-specific knowledge and high-quality text, we curate diverse sources: public financial documents from SEC EDGAR (U.S. Securities and Exchange Commission, 2024), multilingual encyclopedic articles from MegaWika (Barham et al., 2023) and TxT360 (Tang et al., 2024), scientific papers from arXiv (Clement et al., 2019) and peS2o (Soldaini & Lo, 2023), medical data from Pubmed (National Library of Medicine (U.S.), 1996), code from The Stack v1 (Kocetkov et al., 2022b), patents from the USPTO database (United States Patent and Trademark Office, 2024) and EuroPat (Heafield et al., 2022), mathematical problems from Deepmind Math (Saxton et al., 2019), and video transcripts from both VALID (Nguyen et al., 2024) and the YouTube Commons corpus (Langlais, 2024), news and law data from the Open License Corpus (Min et al., 2024).

We source 12.6% of our dataset from **The Stack v1**, a permissive-first, risk-mitigated code dataset governed by the OpenRAIL-M license. We discuss its permissiveness situation in Appendix H.

### 2.1.3 INSTRUCTION AND REASONING DATASETS

To enhance instruction-following and reasoning, we follow Abdin et al. (2024) by including considerable synthetic and web-rewrite data. We extensively use fully and partially synthetic data — all generated from permissive or public-domain seed data using models under permissive licenses.

**General Instruction Following**   We include a strong instruction-following baseline with the Magpie Collection (Xu et al., 2024a), its derivatives (e.g., Magpie-Phi3-Pro). This is augmented with preference data from UltraFeedback (Cui et al., 2023) and NVIDIA's SFT data blend NVIDIA (2024), which contains a curated mixture of permissively licensed subsets from public datasets, including OASST (Köpf et al., 2023), CodeContests (Li et al., 2022), FLAN (Chung et al., 2022), OpenPlatypus (Lee, 2023), and the training split of GSM8K (Cobbe et al., 2021b). Additionally, we augment the P3 (Sanh et al., 2021) dataset with a few-shot and multiple-choice format.

**Reasoning**   To improve reasoning, we incorporate general corpora such as Glaive-AI Reasoning (AI, 2023) and OpenThoughts (Guha et al., 2025) as well as domain-specific datasets: the legal dataset CaseHOLD (Zheng et al., 2021), scientific Q&A from the OpenScience collection (NVIDIA Corporation, 2025), and agent-focused instructions from OpenManus-RL (CharlieDreemur, 2024).

**Mathematics and Coding**   To strengthen quantitative reasoning, we combine our internally developed synthetic Math Word Problems dataset (Appendix F) with established datasets like Meta-MathQA (Yu et al., 2023) and DM-Math (Saxton et al., 2019), further enriched with large-scale math instruction sets, including OpenMathInstruct-2 (Toshniwal et al., 2024), DART-MATH (Ge et al., 2024), Nemo-Math (Mahabadi et al., 2025), and Prism-Math (NVIDIA, 2025). For coding, we combine the Ling Coder collection Codefuse & Team (2025) with executable instructions from the StarCoder dataset Kocetkov et al. (2022b) to target a wide range of software engineering tasks.

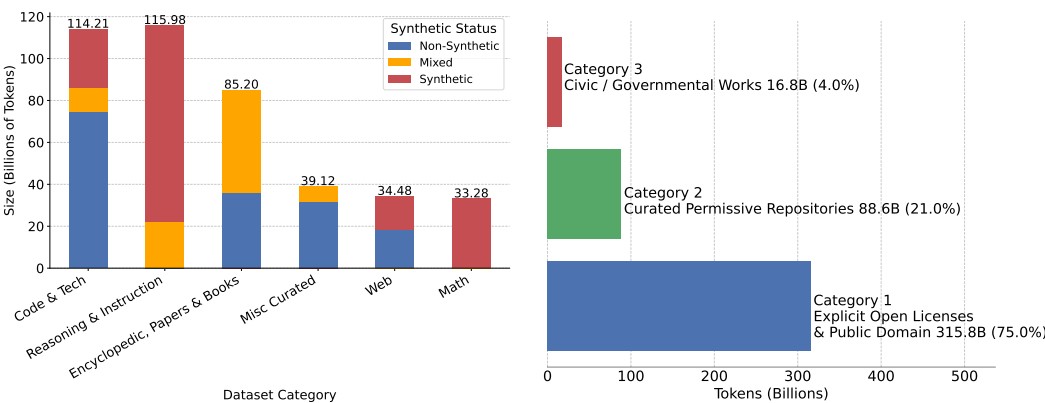

(a) **MixtureVitae** composition by origin (total token counts at the top in billions). Each bar represents one of the six primary content domains (as in Figure 1a), segmented by source type: **Non-Synthetic** (organic human-written text and code), **Mixed** (sources with partial synthetic data), and **Synthetic** (data generated by permissive models from permissive seeds).

(b) Legal provenance and risk-mitigation tiers of the **MixtureVitae** corpus. The dataset is segmented into its three constituent legal categories, with all sources falling into a permissive-first or risk-mitigated tier. Token counts (billions) and total corpus percentages are shown for each category.

Figure 2: Composition and provenance of **MixtureVitae**: **(a)** Synthetic-status distribution across the six content domains, **(b)** licensing tiers and risk posture for the corpus.

### 2.1.4 LICENSING TIERS AND RISK PROFILES

To make the provenance and legal footing of **MixtureVitae** transparent, we conceptualize all dataset components into *tiers* based on license type and expected risk profile (see Figure 2b and Table 13).[1]

**Tier 1 — Explicit Open Licenses & Public Domain.** This tier encompasses text and code under clear permissive licenses (e.g., CC0, CC-BY, Apache 2.0, MIT, BSD, a permissive subset of P3) or in the public domain, such as encyclopedic resources, scientific papers, and portions of curated math corpora. Because licenses are explicit and permissive, the legal risk of reuse is minimal. This tier also includes synthetic data generated from permissively licensed models and seed data.

**Tier 2 — Curated Permissive Corpora with Upstream Opacity.**

- (a) **Permissive Corpora With Partial or Unverified Provenance.** This subset includes resources such as THE STACK V1 and Wikipedia-derived corpora. The released dataset all carries a permissive license, and curators apply filters (e.g., repository-level license heuristics). However, because provenance is only partially tracked at the file or example level, there remains some residual uncertainty about the licensing status of individual items, hence its separation from Tier 1. This Tier also includes datasets that have no license, but the underlying data is public domain or permissive and requiring the same license as the upstream data, or where the data is solely obtained synthetically from a model that is permissively licensed.

- (b) **Synthetic Data with Non-Permissive or Unverifiable Generators or Seeds.** This tier contains datasets that are themselves permissively licensed (e.g., Apache/MIT/CC-BY), but where either (i) the generator model used to create the synthetic data operates under a more restrictive license (e.g., Llama-3 community license, OpenAI API terms), or (ii) the seed data contains slices whose provenance cannot be fully audited (e.g., partially opaque community mixtures). These datasets constitute only ≈4% of **MixtureVitae** and are isolated for transparency so that users who require a strictly permissive generator and seed provenance can exclude them (more detail in Table 13).

**Tier 3 — Civic / Governmental Works.** This tier includes materials that are either statutory public domain (e.g., U.S. federal works) or under a strong public-purpose rationale for reuse (e.g., government websites, regulatory notices). While not always explicitly licensed, such work—typically created for dissemination—is widely recognized as low-risk for inclusion. Filtering with copyright keyword checks further reduces the possibility of inadvertently including restricted content.

### 2.2 DATA PROCESSING PIPELINE

To transform the raw data sources into a high-quality and permissively licensed pretraining corpus, we develop a multistage data processing pipeline. Our curation pipeline includes the following stages: ensuring permissive licensing, filtering for CSAM and offensive language, improving overall content quality, and reducing data redundancy. The following sections detail each component.

### 2.2.1 PERMISSIVENESS FILTERING

In contrast to standard data pipelines that rely on the retroactive negative filtering of broad web scrapes (e.g., Fan et al., 2025), we employ a **positive inclusion** strategy for web data. Rather than ingesting broad web dumps and filtering post-hoc, we positively select sources based on auditable permissive status. Specifically, Specifically, we (i) apply an explicit allowlist of governmental and international domains (Appendix I.1), (ii) curate a set of websites with known permissive licenses (Appendix I.2), and (iii) expand this set with risk-mitigated documents by searching for permissive license keywords (e.g., "CC-BY-SA"), excluding documents with restrictive terms (e.g., "all rights

---

[1] The high-level groupings presented in this section (e.g., "Code & Tech", "Reasoning") and the shard breakdowns in the Appendices are primarily organizational abstractions for visualization and provenance tracking. In practice, the actual training data construction follows a granular *domain-aware mixing strategy* (detailed in Section 2.2.5), where documents are clustered by base URL or provenance to preserve domain coherence per sample, rather than strictly sampling from rigid high-level partitions.

reserved"). This upfront design minimizes the risk of including paywalled or opted-out content (e.g., commercial news). We justify the inclusion of governmental works under a strong fair-use rationale, considering their public purpose, content type, and minimal market impact (Appendix G).

### 2.2.2 SAFETY FILTERING

We remove obscene, adult and CSAM-related content with keyword-based blocklists adapted from prior work (Laurençon et al., 2022; Nakamura et al., 2025). For Wikipedia-based documents, we remove articles about films, sporting events, and biographies of living persons in English with applied targeted filtering, to minimize memorization of facts about people, in case of objection to incorrect facts about people being generated by models trained on MixtureVitae. Besides dataset-level filters, we also evaluate the final model's safety profile via standard red-teaming (Appendix C.3).

### 2.2.3 QUALITY FILTERING

Per standard practices (Raffel et al., 2020), we remove documents with base64-encoded text (which can disrupt training) and duplicative headers and footers (e.g., "Home | Search") from FineFineWeb.

### 2.2.4 DEDUPLICATION

Informed by recent findings in large-scale data curation, our deduplication strategy prioritizes diversity over purity. While removing exact repetitions mitigates harmful memorization (Lee et al., 2022), prior research finds that aggressive, global near-duplicate removal can be detrimental. For example, the creators of the **FineWeb-Edu** dataset (Lozhkov et al., 2024a) reported *worsened* model performance by global fuzzy deduplication, postulating that it removed "too much quality data."

Therefore, we adopt a local-only approach. We first apply **intra-dataset deduplication** using prefix-based exact matching to remove verbatim boilerplate text (Lee et al., 2022). We **intentionally avoid full, cross-dataset fuzzy deduplication** to preserve near-duplicates (e.g., Wikipedia articles with different formatting across sources). We posit that doing so retains **"stylistic and domain diversity,"** a factor shown to be helpful for model generalization (Chen et al., 2024).

### 2.2.5 TRAINING EXAMPLE CURATION

Our process for creating training examples involves several stages:

1. **Heuristic Cleaning:** We remove boilerplate content by eliminating repetitive n-gram prefixes and suffixes, following standard web data cleaning pipelines (Raffel et al., 2020).

2. **Fine-grained Deduplication:** To enhance data quality, we segment documents into sentences and remove duplicate sentences within each document. Documents with high internal repetition (sentence duplication rate > 75%) are discarded entirely, as this has been shown to improve model performance (Lee et al., 2022).

3. **Domain-Aware Mixing:** To construct the final training examples, we employ a domain-aware data mixing strategy (Xie et al., 2023). Documents are clustered by their base URL (a proxy for domain), and sentences are concatenated first within their original document, then packed with other documents from the same cluster.

### 2.2.6 ADDITIONAL FILTERING FOR SYNTHETIC DATASETS

To ensure that the synthetic subsets of MixtureVitae adhere to our permissive-first, risk-mitigated approach, we prioritize data originating from seeds that are sourced from permissive sources and generated with models that are themselves permissively licensed. A small portion (≈4%) of MixtureVitae originates from sources with restricted, mixed, or opaque provenance and is isolated into Tier 2(b), as detailed in Appendix J and Table 13.

## 3 EXPERIMENTS

### 3.1 EXPERIMENTAL SETUP

To empirically validate the quality of the MixtureVitae pretraining dataset, we conduct a large-scale comparative study against a selection of prominent open pretraining datasets. We isolate the impact of the dataset on downstream performance using the **open-sci-ref** training procedure (Nezhurina et al., 2025), which enables systematic control of factors affecting benchmark scores. As in open-sci-ref, we fix the model architecture (Table 3, sizes: 0.13B, 0.4B, 1.3B, 1.7B) and training hyperparameters (Table 4), varying only the dataset. This design ensures that any performance difference can be attributed solely to the dataset.

Also, following the numbers given in open-sci-ref, we train each model on two token budgets: 50B and 300B, to analyze scaling effects. Conducting separate training runs on each budget, rather than using intermediate checkpoints, thus ensuring a consistent data distribution and allowing for proper optimization of learning rate schedules for each specific token budget (Hoffmann et al., 2022). This follows standard practice: Data mixtures effective at small token budgets may not generalize to larger ones (Albalak et al., 2023).

To guard against test-set leakage, we also perform a large-scale 13-gram decontamination analysis and re-evaluation; Section 3.5 and Appendix D detail this procedure.

Within this controlled evaluation framework, we compare MixtureVitae with the set of public baselines evaluated in open-sci-ref, with the addition of a representative selection of permissively licensed datasets. As detailed in Table 2, the comparison set includes two groups:

- **Non-Permissive/Mixed-License Baselines.** C4 (Raffel et al., 2020), The Pile (Gao et al., 2020), SlimPajama (Shen et al., 2024), FineWeb-Edu (Lozhkov et al., 2024a), Nemotron-CC-HQ (Su et al., 2025), DCLM-baseline (Li et al., 2024), HPLT Monolingual Datasets v2.0 (Burchell et al., 2025);

- **Permissive Baselines.** CommonCorpus and its English subset (Pleias, 2023), as well as Comma-0.1 (Kandpal et al., 2025).

All datasets are tokenized using the GPT-NeoX-20B tokenizer (Black et al., 2022), resulting in a vocabulary size of 50,304. The models are trained using Megatron-LM (Shoeybi et al., 2020), and the evaluations are performed using LM Evaluation Harness (Gao et al., 2024).

Model performance is evaluated on recognized downstream task benchmarks: MMLU (Hendrycks et al., 2021), COPA (Gordon et al., 2012), LAMBADA (Paperno et al., 2016), OpenBookQA (Mihaylov et al., 2018), Winogrande (Sakaguchi et al., 2020), ARC (Challenge and Easy) (Clark et al., 2018), BoolQ (Clark et al., 2019), HellaSwag (Zellers et al., 2019), and PIQA (Bisk et al., 2020).

To ensure evaluation integrity, we perform a comprehensive decontamination analysis against all benchmark test sets, with full details and case studies provided in Appendix D.

### 3.2 EXPERIMENT RESULTS

At a 300B-token budget, MixtureVitae shows strong scaling behavior when compared to the reference permissive dataset and is almost comparable to the non-permissive datasets (Figure 3).

**Overall average performance.** MixtureVitae outperforms all permissive dataset baselines by a significant margin, with gaps widening considerably for larger model sizes, in terms of average performance across all 10 tasks (see Figure 3a). Non-permissive datasets, particularly Nemotron-CC-HQ and DCLM, still achieve the highest overall performance. Approaching the 300B token budget, MixtureVitae catches up to FineWeb-Edu and DCLM. More importantly, while the top-performing models are still trained on non-permissive datasets like Nemotron-CC-HQ and DCLM, our results demonstrate that this performance gap is no longer an inevitability. MixtureVitae proves that a dataset built on a fully permissive, risk-mitigated foundation can achieve highly competitive results—significantly outperforming all other permissive baselines and landing within a small, practical margin of top-tier, legally-ambiguous corpora. This finding directly challenges the prevailing

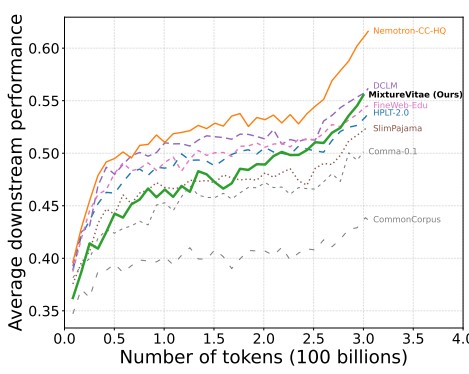 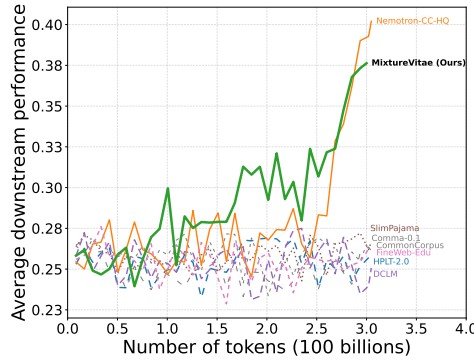

(a) Average performance on all 10 evaluated tasks.     (b) Performance on MMLU.

Figure 3: Performance comparison of pretraining datasets for a 1.7B-parameter model trained up to a 300B token budget, showing downstream accuracy as a function of the number of training tokens.

assumption that reliance on high-risk, indiscriminately scraped copyrighted data is a prerequisite for training capable LLMs. Our findings also hold at the 50B token budget scale (Appendix C.2).

**Performance on MMLU.**   As demonstrated in figure 3b, **MixtureVitae** performs particularly well relative to others on **MMLU**, where most baselines fall within near random chance throughout. Among all the baselines, only Nemotron-CC-HQ catches up to **MixtureVitae** at around 260B and overtakes it past that point.

**Performance on single tasks.**   We show performance on each task separately in the appendices (Fig. 5). **MixtureVitae** outperforms other permissive datasets on MMLU, Arc Challenge, Arc Easy and BoolQ, while closely matching DCLM and FineWeb-Edu. On PIQA, HellaSwag, Winogrande, OpenBookQA, **MixtureVitae** is on par with Comma-0.1, while both are behind non-permissive datasets. Lambada is the only task where **MixtureVitae** falls behind Comma-0.1. We thus observe **MixtureVitae** to be particularly strong on reasoning-related tasks.

### 3.3   Results on Problem Solving and Instruction-Based Downstream Tasks

To further demonstrate the performance of the **MixtureVitae** dataset, we evaluate the model on a set of math, code, and instruction benchmarks: GSM8k (Cobbe et al., 2021b), MBPP (Chen et al., 2021), IF-Eval (Zhou et al., 2023). Our evaluation uses the final 1.7B model checkpoints after training for 300B tokens using the `open-sci-ref` protocol (exact evaluation setup in Table 6).

Unlike traditional web-only baselines (e.g., C4, FineWeb), **MixtureVitae** utilizes a *reasoning and instruction-heavy* pretraining mixture. Compared against base models for architectural consistency, this front-loading strategy yields capabilities typically associated with post-training, demonstrating a more token-efficient path to reasoning competence than relying on post-training alone.

The results (Table 1) show a dramatic difference on math (GSM8K) and coding (HumanEval, MBPP). **MixtureVitae** achieves scores of **0.53**, **0.32**, and **0.38**, respectively. This performance is considerably stronger than any other dataset, all of which remain near random performance on GSM8K (0.02-0.06) and cap at 0.13 on HumanEval and 0.22 on MBPP. Most notably, our base model outperforms the post-trained SmolLM2-1.7B-Instruct (Allal et al., 2025) model on GSM8K, HumanEval, and MBPP — despite the latter being trained on $\approx$11T tokens (over $36\times$ our budget).

### 3.4   Role of Instruction and Reasoning Data

**MixtureVitae**'s performance is enhanced by the large proportion of reasoning and instruction data, as demonstrated in the ablation study in Appendix E. Removing this subset ("w/o Instructions" in Figure 10) causes a substantial degradation across tasks—far larger than the impact of removing the

Table 1: **Performance on math, code, and instruction-following tasks for 1.7B models.** We compare MixtureVitae—trained on a reasoning- and instruction-heavy, permissive-first mixture—against standard `open-sci-ref` baselines trained on predominantly web-based corpora. MixtureVitae shows a substantial lead in math and code tasks. Notably, the 1.7B MixtureVitae base model exceeds **SmolLM2-1.7B-Instruct** on GSM8K, HumanEval, and MBPP despite training on 300B rather than ≈11T tokens.

| Training Dataset | Tokens | IF-Eval | GSM8K | HumanEval | MBPP | Average |
|---|---|---|---|---|---|---|
| *Models Trained with* `open-sci-ref` *for 300B Tokens* | | | | | | |
| **MixtureVitae** | 300B | 0.19 | **0.53** | **0.32** | **0.38** | **0.36** |
| Comma-0.1 | 300B | 0.19 | 0.06 | 0.13 | 0.22 | 0.15 |
| CommonCorpus | 300B | 0.13 | 0.02 | 0.05 | 0.05 | 0.06 |
| C4 | 300B | 0.20 | 0.02 | 0.00 | 0.00 | 0.06 |
| SlimPajama | 300B | 0.14 | 0.02 | 0.05 | 0.00 | 0.05 |
| HPLT-2.0 | 300B | 0.17 | 0.02 | 0.00 | 0.00 | 0.05 |
| DCLM | 300B | 0.13 | 0.02 | 0.01 | 0.01 | 0.04 |
| Nemotron-CC-HQ | 300B | 0.09 | 0.03 | 0.02 | 0.00 | 0.03 |
| *Models Trained with* `open-sci-ref` *for 1T Tokens* | | | | | | |
| FineWeb-Edu | 1T | 0.20 | 0.03 | 0.00 | 0.00 | 0.06 |
| Nemotron-CC-HQ | 1T | 0.13 | 0.03 | 0.01 | 0.04 | 0.05 |
| DCLM | 1T | 0.15 | 0.03 | 0.00 | 0.01 | 0.05 |
| *Other Models* | | | | | | |
| SmolLM2-1.7B | 11T | 0.18 | 0.31 | 0.01 | 0.35 | 0.21 |
| SmolLM2-1.7B-Instruct | 11T | **0.28** | 0.37 | 0.28 | 0.37 | 0.33 |

web component. This effect validates and extends the findings of Phi-4 (Abdin et al., 2024), showing that a permissive-first, risk-mitigated, and reasoning-heavy mixture can substitute vast quantities of generic web text, particularly under constrained token budgets.

### 3.5 CONTAMINATION AND LEAKAGE

To rule out test-set leakage as an alternative explanation for these gains, we perform a 13-gram exact-match decontamination sweep between MixtureVitae and all benchmarks (Appendix D.3). Document-level overlap is negligible for most tasks (e.g., at or below 0.0003% for ARC, HellaSwag, LAMBADA, OpenBookQA, and PIQA; see Table 8); contamination rates are modest for MMLU and BoolQ; for code benchmarks such as HumanEval and MBPP, contamination rates are higher but still small. We then (i) re-evaluate all models on decontaminated test sets with all overlapping items removed and (ii) retrain a 1.7B model, removing the shards responsible for the majority of the contamination signal. In both cases, MixtureVitae's performance remains essentially unchanged (Table 9 and Figure 8), confirming that our results are not an artifact of test-set leakage.

## 4 RELATED WORK

LLM development is intrinsically linked to the scale and quality of pretraining datasets, which have become larger, more diverse, with a growing emphasis on provenance and licensing recently.

### 4.1 PIONEERING LARGE-SCALE DATASETS

Early large-scale text corpora for language modeling often rely on web-crawled data for scale. C4 (Raffel et al., 2020), derived from Common Crawl, is instrumental in training the T5 model, setting standards for large-scale data cleaning and deduplication. Gao et al. (2020) then introduce The Pile, demonstrating the benefit of a more varied data mixture on model generalization and downstream performance. Similarly, ROOTS (Laurençon et al., 2022) supports the training of the BLOOM model with its 498 Common Crawl multilingual scrapes. While foundational, these datasets often have complex or unspecified licenses, mixing permissive data with content of unknown or non-commercial licensing, creating potential legal risks for commercial applications.

## 4.2 OPEN AND REPRODUCIBLE DATASETS

Amidst many proprietary "black box" datasets, the community has pushed for more openness and reproducibility, moving toward permissive datasets that are also performant, e.g., RedPajama-1T (Weber et al., 2024) and its processing recipes (Touvron et al., 2023), Dolma (Soldaini et al., 2024) and its open-source curation toolkits, SILO (Min et al., 2024). Our work joins this effort, contributing a new risk-mitigated dataset featuring explicit consideration for the underlying copyright.

## 4.3 PERMISSIVELY LICENSED AND SYNTHETIC DATA

Growing awareness of copyright and data ownership has spurred interest in datasets built solely from permissively licensed materials. The Stack (Kocetkov et al., 2022a) curates such data for code-generation models, but creating a large, diverse, and high-quality corpus for natural language from exclusively permissive sources remains a challenge. Recent efforts like Common Corpus (Langlais et al., 2025) and The Common Pile (Kandpal et al., 2025) advance the creation of large-scale corpora of permissively licensed and public-domain text. While foundational, our experiments (Section 3) show that models trained on them can lag in complex reasoning, math, and instruction following, suggesting that strictly permissive human text alone is insufficient to instill these advanced skills.

With this scarcity of high-quality reasoning and instruction data, researchers have turned to synthetic data. Alpaca (Taori et al., 2023) and OpenMathInstruct-1 (Toshniwal et al., 2024) use instructional data for fine-tuning. Phi4 proposes using synthetic data for reasoning tasks (Abdin et al., 2024). Our work, **MixtureVitae**, extends these trends with a meticulously curated, permissive-first, risk-mitigated dataset augmented with targeted synthetic data, providing a strong, legally considered foundation for LLM training to mitigate copyright risks in many existing corpora.

While both our work and the concurrent Apertus project (Hernández-Cano et al., 2025) value openness and legal safety, they represent distinct, complementary design philosophies. First, regarding scale versus efficiency, Apertus optimizes for breadth, processing 15T tokens across 1800+ languages using retroactive filtering (e.g., `robots.txt`) on large web-scale datasets. In contrast, **MixtureVitae** focuses on data efficiency through a *positive inclusion strategy*, curating sources known to be permissive (e.g., government works, The Stack) and prioritizing English-centric reasoning density. Our results demonstrate that a reasoning-heavy mixture can achieve strong performance on MMLU, GSM8K, and MBPP with roughly 2% of the pretraining token budget of a dataset in the size range of Apertus. Finally, whereas Apertus primarily releases recipes and reconstruction scripts, **MixtureVitae** provides a single, ready-to-use pretraining mixture.

For a comparison of **MixtureVitae** to other permissive and non-permissive baselines, see Table 2.

## 5 DISCUSSION & CONCLUSION

We have introduced **MixtureVitae**, a pretraining corpus serving as a proof-of-concept: Permissively licensed and permissively-sourced synthetic data can achieve high performance. Our results suggest a shift in the **compliance–performance frontier**. **MixtureVitae** demonstrates that capabilities previously associated with mixed-license corpora are reachable with a permissive first, risk-mitigated approach. In our controlled 300B-token experiments, not only does **MixtureVitae** catch up to leading non-permissive baselines like DCLM and FineWeb-Edu, but our 1.7B base model also outperforms the *post-trained* SmolLM2-1.7B-Instruct—a model trained on ≈11T tokens—on GSM8K, HumanEval and MBPP. Building upon the work of Phi-4 (Abdin et al., 2024), this confirms that a high-quality, permissive-first dataset—rich in reasoning and instruction data—can effectively substitute for vast quantities of broadly scraped web text.

Beyond this specific corpus, the three-tier licensing scheme and its shard-level annotations provide a **concrete template for structuring risk-mitigated mixtures in future work**, and **MixtureVitae** as a whole serves as a reusable blueprint for compliant pretraining. We demonstrate a pipeline built on positive-inclusion "pseudo-crawling," tiered provenance tracking, and targeted synthetic generation with audited seeds. As detailed in our scaling outlook (Appendix K), this recipe provides a path to extend compliant pretraining to the multi-trillion-token regime—via subset upsampling, multilingual expansion, and synthetic growth—providing the community with a sustainable alternative to the legal uncertainty of broad web scrapes.

## 6 REPRODUCIBILITY STATEMENT

We release our code at `https://anonymous.4open.science/r/mixturevitae-FEFE`, with a frozen snapshot at commit `6785991a` corresponding to this submission.

### 6.1 DATASET AND CURATION RECIPES

- **Public Release:** The full 422**B** token dataset, along with the 100B and 50B subsets used for scaling ablations experiments, will be made publicly available upon acceptance of this paper.
- **Curation Methodology:**
  - **Dataset Composition** The detailed list of sources and their composition are shown in Figure 4.
  - **Code**: We are including our data curation and math word problem generation scripts with the submission.

### 6.2 TRAINING PROCEDURE

To ensure our experiments are directly comparable and reproducible, we adhered to a controlled, public framework.

- **Framework:** All experiments were conducted using the `open-sci-ref` training procedure (Nezhurina et al., 2025), which standardizes key factors affecting performance.
- **Architectures:** The exact model architectures for all four scales (0.13B, 0.4B, 1.3B, 1.7B) are detailed in Table 3.
- **Hyperparameters:** The complete training schedules and hyperparameters (learning rate, batch size, warmup, etc.) for both the 50B and 300B token budgets are specified in Appendix B.1.
- **Software:** Models were trained using Megatron-LM (Shoeybi et al., 2020) with the GPT-NeoX-20B tokenizer(Black et al., 2022).
- **Code**: We are including our training script with the submission.

### 6.3 EVALUATION AND ANALYSIS

Our evaluation protocol is fully specified to allow for independent verification of our results.

- **Framework:** All general and reasoning task evaluations were performed using the public LM Evaluation Harness (Gao et al., 2024).
- **Settings:** The exact settings for each benchmark, including the number of few-shot examples, are provided in Table 5 and Table 6.
- **Decontamination:** Our 13-gram decontamination protocol is detailed in Appendix D.
- **Code**: We are including our evaluation and decontamination scripts with the submission.

While model checkpoints and training logs are not included in the initial submission due to size and anonymity constraints, we plan to release these upon publication to facilitate future research.

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

# Appendix: MixtureVitae – Open Web-Scale Pretraining Dataset With High Quality Instruction and Reasoning Data Built from Permissive Text Sources

## A   DATASET COMPOSITION AND COMPARISON

This appendix provides a detailed view of the MixtureVitae corpus, both in relation to other datasets and in its internal construction.

Table 2: Comparison of large-scale pretraining datasets, grouped by their licensing philosophy to provide context for our performance results. MixtureVitae is unique in its combination of a risk-mitigated licensing approach and the inclusion of a large subset of reasoning, coding and instruction synthetic data.

| Dataset | Size (Tokens) | Primary Data Types | Licensing Philosophy |
|---|---|---|---|
| *Non-Permissive / Mixed-License Baselines* | | | |
| Nemotron-CC-HQ (Su et al., 2025) | $\approx$ 1.1T | Web, Synthetic | Unspecified |
| DCLM-baseline (Li et al., 2024) | $\approx$ 3.8T | Web, Code, Academic | Mixed / Unspecified |
| FineWeb-Edu (Lozhkov et al., 2024b) | $\approx$ 1.3T | Web (Educational) | Unspecified |
| The Pile (Gao et al., 2020) | $\approx$ 183.28B | Web, Books, Code | Mixed / Unspecified |
| SlimPajama (Shen et al., 2024) | $\approx$ 627B | Web, Books, Code | Mixed / Unspecified |
| C4 (Raffel et al., 2020) | $\approx$ 156B | Web | ODC-BY |
| HPLT-2.0 (eng.) (Burchell et al., 2025) | $\approx$ 2.86T | Web, Books, News | Mixed / Unspecified |
| *Permissive Baselines* | | | |
| CommonCorpus (Pleias, 2023) | $\approx$ 2T | Web, Curated | Strictly Permissive |
| Comma-0.1 (Kandpal et al., 2025) | $\approx$ 1T | Web, Curated | Strictly Permissive |
| KL3M (Bommarito II et al., 2025) | $\approx$ 580B | Web, Curated | Strictly Permissive |
| OLC (Min et al., 2024) | $\approx$ 228B | Web, Curated | Strictly Permissive |
| *Our Contribution* | | | |
| **MixtureVitae** | $\approx$ **422B** | **Web, Curated, Synthetic** | **Permissive-First, Risk-Mitigated** |

**Shard Definitions and Mixing.**   It is important to note that the dataset shards and categories listed in this appendix serve as logical groupings for transparency, licensing audits, and ablation analysis. They do not dictate a rigid sequential training order. As noted in the main text, the physical construction of training batches utilizes domain-aware packing to maximize local coherence, prioritizing the density of reasoning and factual tokens over these high-level taxonomic boundaries.

Table 2 presents a high-level comparison of MixtureVitae against the other prominent pretraining datasets evaluated in our experiments, detailing their respective sizes, primary data types, and licensing philosophies.

Figure 4 presents the detailed composition of the MixtureVitae dataset. The individual components are color-coded by their primary dataset category, as presented in the main text.

- **Code & Tech (Blue):** This domain is anchored by our largest code sources, Stack V1 and Ling-Coder, and supplemented by StackExchange.

- **Reasoning & Instruction (Green):** The largest contributor to this category is Open Thoughts , followed by P3 and NVIDIA OpenScience.

- **Encyclopedic, Papers & Books (Purple):** This category is dominated by Wikipedia, the single largest component in the dataset. It is complemented by large-scale text from PubMed and arXiv.

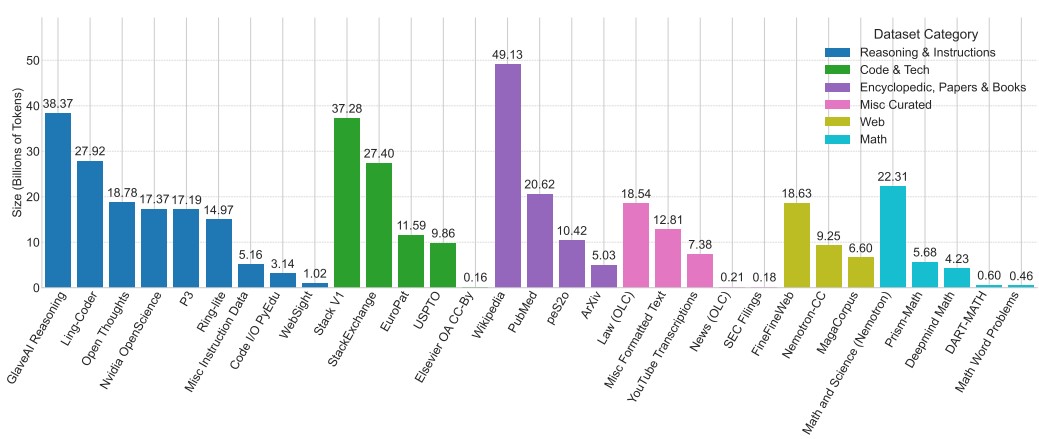

Figure 4: Detailed composition of the **MixtureVitae** dataset.

- **Math (Cyan):** The math component is a diverse mixture of sources, led by the Math and Science (Nemotron) corpus and Prism-Math.

- **Web (Yellow):** Our web data is primarily sourced from corpora such as SEC Filings, Mega-Corpus, and FineFineWeb.

- **Misc Curated (Pink):** This category includes a variety of high-quality curated sources, notably Law (Open License Corpus) and YouTube Transcriptions.

## B  EXPERIMENT SETUP DETAILS

To ensure full reproducibility, this appendix details the complete experimental setup. This includes the model architectures for all scales, the training hyperparameters for both 50B and 300B token budgets, and the specific settings used for all general evaluation benchmarks.

Table 3: **open-sci-ref** (Nezhurina et al., 2025) model architecture and scales. We used tied embedding weights in all experiments.

| Parameters (B) (Non-Emb + Emb) | Layers | Hidden | Heads | FFN Hidden | Memory | FLOPs |
|---|---|---|---|---|---|---|
| $0.1 + 0.03 = 0.13$ | 22 | 512 | 8 | 2256 | $0.89\,\text{GB}$ | $7.8 \times 10^8$ |
| $0.35 + 0.05 = 0.40$ | 22 | 1024 | 16 | 3840 | $2.88\,\text{GB}$ | $2.4 \times 10^9$ |
| $1.21 + 0.10 = 1.31$ | 24 | 2048 | 32 | 5440 | $7.544\,\text{GB}$ | $7.9 \times 10^9$ |
| $1.61 + 0.10 = 1.71$ | 24 | 2048 | 32 | 8192 | $9.884\,\text{GB}$ | $1.0 \times 10^{10}$ |

Table 4: The training schedules used in our experiments.

| Tokens | Global Batch Size (tokens) | Iterations | Learning Rate | Warmup | Cooldown (20%) |
|---|---|---|---|---|---|
| 50B | 4.12M | 11,921 | $4 \times 10^{-3}$ | 1,000 | 2,384 |
| 300B | 4.12M | 72,661 | $4 \times 10^{-3}$ | 25,000 | 14,532 |

### B.1 TRAINING SETUP PARAMETERS

This appendix details the exact model architectures and training hyperparameters used for all experiments, ensuring full reproducibility.

We adopt the standard architectures and scales from the **open-sci-ref** framework to allow for a fair and direct comparison against other published baselines. All models were trained with tied embedding weights.

**Model Architecture** Table 3 defines the four model scales used in our study. The columns are defined as follows:

**Parameters (B) (Non-Emb + Emb)** The total model parameters in billions, separated into **Non-Embedding** (Non-Emb) parameters (the core transformer blocks) and **Embedding** (Emb) parameters (the token lookup tables). As noted in the caption, we used tied embedding weights.

**Layers** The total number of transformer blocks stacked in the model.

**Hidden** The hidden size (or embedding dimension, $d_{\text{model}}$) of the model.

**Heads** The number of attention heads in the multi-head attention mechanism.

**FFN Hidden** The inner dimension of the Feed-Forward Network (FFN) layer within each transformer block.

**Memory** The approximate VRAM required to store the model weights, in bfloat16.

**FLOPs** An approximation of the training compute cost using the **6N** rule: a standard estimate for a transformer's forward-and-backward pass, where **N** is the number of *non-embedding* parameters (Kaplan et al., 2020).

**Training Schedules** Table 4 defines the training hyperparameters for our two main experimental runs (50B and 300B tokens). We use a single stage training with no post-training.

**Tokens** The total number of tokens in the training run.

**Global Batch Size (tokens)** The total number of tokens processed in a single training step (i.e., one gradient update) across all GPUs.

**Iterations** The total number of training steps.

**Learning Rate** The peak learning rate used during training.

**Warmup** The number of initial *iterations* (steps) over which the learning rate linearly increases from 0 to its peak value.

**Cooldown (20%)** The number of final *iterations* (the last 20% of training) over which the learning rate decays to zero.

### B.2 EVALUATION SETTINGS

We used the `lm-evaluation-harness` (Gao et al., 2024) for all general evaluations. The specific tasks and few-shot counts are detailed in Table 5. The settings for the reasoning tasks (e.g., GSM8K, IFEval) are listed separately in Table6.

## C ADDITIONAL EXPERIMENTS

This appendix provides additional experimental results to supplement the findings presented in the main paper. We offer a more granular breakdown of the 300B token experiment, analyze performance at a smaller 50B token scale to assess the generalization of our results, and report the results of a model red-teaming analysis to evaluate the model's safety profile.

### C.1 300B EXPERIMENT - DETAILED RESULTS

The detailed results for each evaluated task are given in Figure 5.

Table 5: General evaluation benchmark settings. All tasks use Accuracy as the primary metric.

| Task | Citation | # of Shots |
|------|----------|------------|
| MMLU | Hendrycks et al. (2021) | 5 |
| HellaSwag | Zellers et al. (2019) | 10 |
| ARC-Challenge | Clark et al. (2018) | 10 |
| ARC-Easy | Clark et al. (2018) | 10 |
| PIQA | Bisk et al. (2020) | 10 |
| BoolQ | Clark et al. (2019) | 10 |
| Winogrande | Sakaguchi et al. (2020) | 0 |
| OpenBookQA | Mihaylov et al. (2018) | 0 |
| COPA | Gordon et al. (2012) | 0 |
| LAMBADA | Paperno et al. (2016) | 0 |

Table 6: Evaluation settings for reasoning tasks. All tasks use Accuracy as the primary metric. To execute the evaluation, we used LM Evaluation Harness Gao et al. (2024).

| Task | Citation | # of Shots |
|------|----------|------------|
| GSM8k | Cobbe et al. (2021b) | 4 |
| IFEval | Zhou et al. (2023) | 0 |
| MBPP | Austin et al. (2021) | 4 |

### C.2 PERFORMANCE AT 50B TOKENS SCALE.

To assess performance on a smaller reference tokens scale, we also evaluated models trained on a 50B token subset of each dataset. The results, shown in Figure 6 and Figure 7, indicate that the advantages of MixtureVitae manifest already at the smaller token scales. Figure 6 shows that MixtureVitae establishes a consistent performance lead over other permissive datasets within the first 50B tokens, especially at the 1.3B and 1.7B model scales. The per-benchmark analysis further reinforces this finding (see Figure 7). On MMLU, MixtureVitae is the only permissive dataset to show a significant learning signal early in training, demonstrating that its composition provides immediate benefits, which might be both due to knowledge rich and instruction like content. Arguably, this suggests that the reasoning capability shown by MixtureVitae is not a late-stage phenomenon but rather an indication of efficient instillation from the early stages of training. This strong initial performance underscores the learning efficiency of MixtureVitae, making it a compelling choice for achieving high performance with less computational cost.

### C.3 MODEL RED TEAMING

To evaluate the safety of the model trained on MixtureVitae for 300B tokens, we performed a red-teaming analysis to measure the Attack Success Rate (ASR) against three standard benchmarks: **ToxiGen** (Hartvigsen et al., 2022), **Do-Not-Answer** (Wang et al., 2023), and **AdvBench** (Zou et al., 2023). The results (Table 7) shows that our model is competitive with the baselines.

The model responses were evaluated using two safety classifiers: (i) **Llama Guard-8B** (Inan et al., 2023), used to evaluate the **Do-Not-Answer** and **AdvBench** datasets, while (ii) the **toxigen_roberta** classifier (Logacheva et al., 2022) was used for the **ToxiGen** benchmark.

Table 7: Attack Success Rate in %, lower is better. All models are trained with the same **open-sci-ref** procedure (300B-token budget) while varying only the pretraining dataset.

| Benchmark | MixtureVitae | Comma | CommonCorpus-Eng | Nemotron-HQ-CC |
|-----------|--------------|-------|------------------|----------------|
| ToxiGen | 8.07 | 9.04 | 12.77 | 10.21 |
| Do-Not-Answer | 28.22 | 24.71 | 21.62 | 20.98 |
| AdvBench | 86.92 | 92.12 | 70.58 | 85.77 |

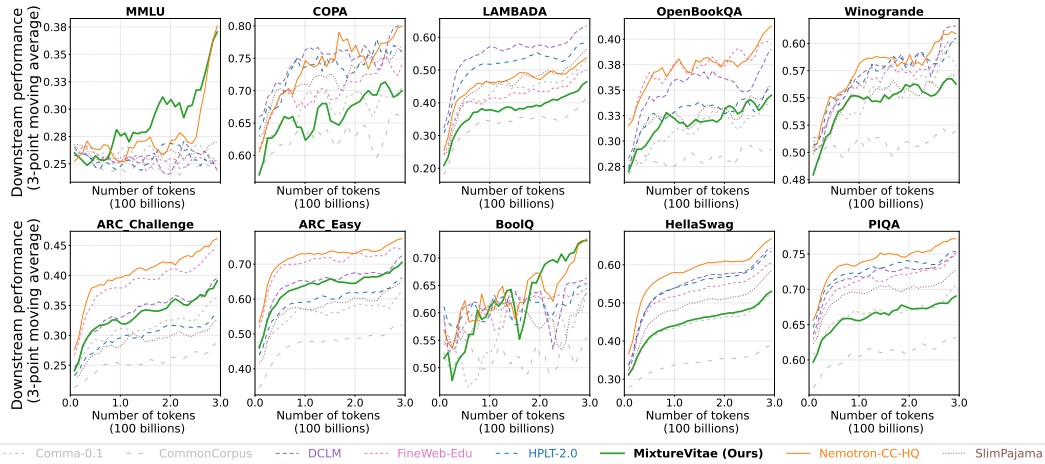

Figure 5: Comparing performance of 1.7B model trained on MixtureVitae and baseline datasets for a 300B token budget. While some evaluations provide clear dataset rankings (e.g. ARC, Hellaswag, Lambada), others do not provide a good signal for dataset comparison, on an individual basis.

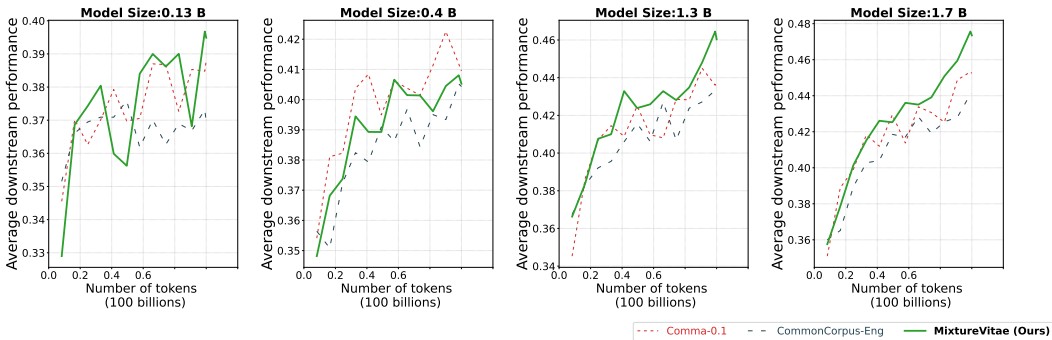

Figure 6: Average performance of permissive datasets after 50B training tokens. MixtureVitae shows an early and consistent lead at larger model scales.

## D CONTAMINATION ANALYSIS

### D.1 CONTAMINATION DETECTION PROTOCOL

To ensure the integrity of our evaluation, we implemented a comprehensive decontamination protocol to measure the overlap between our training dataset and all evaluation benchmarks we report results on. This protocol consists of three main stages: Index Construction, Dataset Scanning, and Leakage Reporting.

#### D.1.1 INDEX CONSTRUCTION

The first stage creates a compact, indexed set of unique n-grams from all benchmark evaluation data.

1. **Text Normalization:** All text from the benchmarks is processed through a normalization pipeline, similar to Laurençon et al. (2022): (1) Unicode normalization (NFKC), (2) conversion to lowercase, (3) tokenization, and (4) removal of a predefined list of common English stop words. This procedure focuses the resulting n-grams on substantive content.

2. **N-gramming and Filtering:** We generate 13-grams, a common n-gram size for this task Brown et al. (2020); Gao et al. (2020) from the normalized token lists. As in Laurençon et al. (2022), a set of regular expressions is used to filter out common boilerplate, exam instructions, and formatting artifacts.

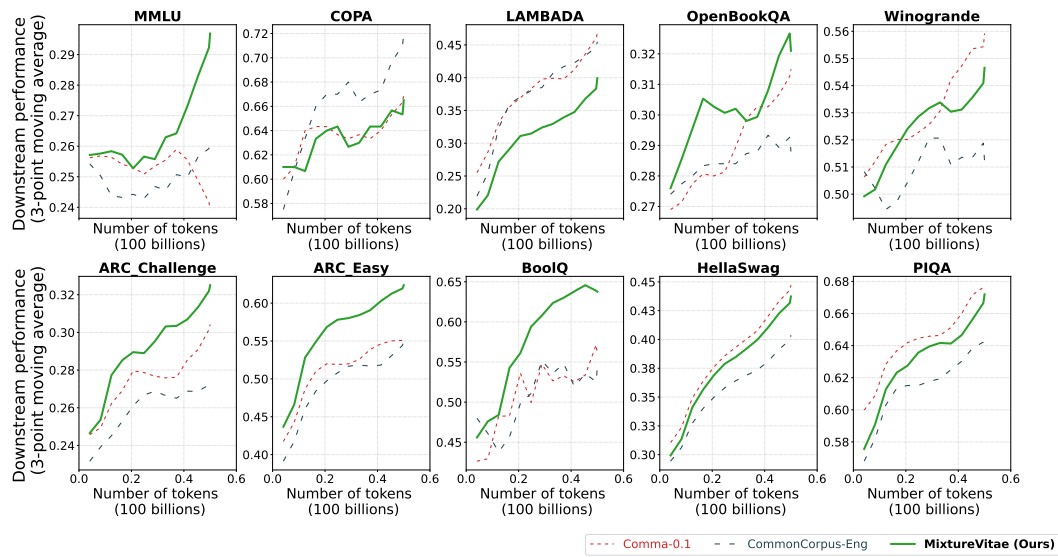

Figure 7: Per-benchmark performance of permissive datasets after 50B training tokens. **MixtureVitae**'s advantage on MMLU is apparent even at this early stage.

3. **Train/Test De-duplication:** as in Gao et al. (2020), we compute the set of all 13-gram hashes from the `train` split and subtract this set from the 13-gram hashes generated from the `test` split. This ensures our index only contains n-grams that are unique to the evaluation set.

### D.1.2 DATASET SCANNING

The second stage analyzes the target training dataset against the generated index.

1. **Document Processing:** Each document in the training dataset is processed using the *exact same* normalization, 13-gramming, and hashing pipeline used for index construction.

2. **Contamination Criteria:** A document is flagged as "contaminated" if it meets two criteria, based on the set intersection of its n-gram hashes with the benchmark index:

   - **Minimum Hits:** The number of distinct matching n-grams is $\geq 3$.
   - **Minimum Coverage:** As proposed in Rae et al. (2021), the coverage of matching n-grams is $\geq 0.1\%$. Coverage is defined as:

$$\text{Coverage} = \frac{\text{distinct\_hits}}{\text{total\_unique\_13grams\_in\_doc}}$$

### D.1.3 LEAKAGE REPORTING

The final stage aggregates the scan results into a summary report.

1. **Numerator (Leaked N-grams):** The procedure aggregates the reports from all scanned partitions. It performs a global *set union* to find all unique n-gram hashes that were found *at least once* in the target dataset, aggregated by benchmark source. This provides the unique_ngrams_leaked count for each benchmark.

2. **Denominator (Total N-grams):** The procedure retrieves the pre-computed metadata to obtain the total unique n-gram count for each benchmark.

3. **Final Metric:** As proposed in Touvron et al. (2023), the **Leak Percentage** for each benchmark is then calculated as:

$$\text{Leak Percentage} = \frac{\text{unique\_ngrams\_leaked}_{\text{benchmark}}}{\text{total\_unique\_ngrams\_in\_index}_{\text{benchmark}}} \times 100$$

## D.2 CONTAMINATION REPORT

We executed our 13-gram contamination scan across the entire 345,697,271 documents of the **MixtureVitae** dataset. The global summary of contaminated documents per benchmark is presented in Table 8.

The results confirm that for the vast majority of benchmarks—including ARC, HellaSwag, LAMBADA, OpenBookQA, and PIQA—the document-level contamination rate is negligible (at or below 0.0003%), strongly validating the integrity of our evaluation on these tasks.

The scan did, however, flag a minor overlap for MMLU (0.0098%) and BoolQ (0.0087%), and a more significant overlap for our key code benchmarks: HumanEval (0.0988%) and MBPP (0.0878%). This overlap in code benchmarks is a known challenge when including large-scale permissive code corpora like The Stack, which may naturally contain snippets of common coding problems (a "source overlap" rather than a direct "test-set leak").

To ensure this overlap did not artificially inflate our model's strong performance on these key tasks, we conducted case studies for the benchmarks with the highest overlap. This analysis is detailed in the following section (Appendix D.3).

Table 8: Global contamination summary by document count, based on a 13-gram overlap scan. This table shows the total number of documents in **MixtureVitae** that contained at least one overlapping n-gram from each benchmark's test set. The total documents in **MixtureVitae** is 345,697,271 and the overall contamination rate is 0.1420%.

| Benchmark | Contaminated Docs | Contamination Rate (%) |
|---|---|---|
| ALERT | 12 | 0.0000% |
| ARC | 17 | 0.0000% |
| BoolQ | 30,144 | 0.0087% |
| GPQA | 1,077 | 0.0003% |
| GSM8K | 230 | 0.0001% |
| HellaSwag | 186 | 0.0001% |
| HumanEval | 341,554 | 0.0988% |
| IfEval | 756 | 0.0002% |
| LAMBADA | 23 | 0.0000% |
| MBPP | 303,558 | 0.0878% |
| MMLU | 33,922 | 0.0098% |
| OpenBookQA | 60 | 0.0000% |
| PIQA | 5 | 0.0000% |
| SimpleQA | 98 | 0.0000% |

## D.3 DECONTAMINATED TEST SET PERFORMANCE

To understand how test data leakage affects final performance on downstream tasks, we conducted the following experiment on all models and benchmarks reported in Section 3.3:

1. Identify problems from the test set that have at least one 13-gram match in the training dataset.

2. Evaluate the model on a decontaminated benchmark version obtained by removing problems that were identified.

As we can see from Table 9, the performance of the evaluated models is consistent between the original and decontaminated versions, aside from some upward bias in the decontaminated versions of MBPP+ and IFEval. Crucially, the result for the model trained on **MixtureVitae** was not affected by the strict decontamination procedure applied to the benchmarks. This rules out the possibility that the strong performance of **MixtureVitae** on math and coding is due to benchmark leakage.

Table 9: Validating math, code, and instruction performance by comparing original (Orig) vs. decontaminated (Decont) test sets for 1.7B models trained for 300B tokens. MixtureVitae's high scores are shown to be genuine, as performance is maintained after removing all overlapping test items. This confirms the model's capabilities are not an artifact of test set leakage.

| Training Dataset | GSM8K | | GSM8K-CoT | | MBPP | | MBPP+ | | IFEval | |
|---|---|---|---|---|---|---|---|---|---|---|
| | Orig | Decont | Orig | Decont | Orig | Decont | Orig | Decont | Orig | Decont |
| **MixtureVitae** | 0.53 | 0.54 | 0.50 | 0.50 | 0.38 | 0.38 | 0.55 | 0.59 | 0.19 | 0.23 |
| SmolLM2 | 0.30 | 0.30 | 0.28 | 0.29 | 0.35 | 0.35 | 0.48 | 0.48 | 0.17 | 0.20 |
| Comma-0.1 | 0.06 | 0.06 | 0.09 | 0.09 | 0.21 | 0.23 | 0.28 | 0.28 | 0.18 | 0.20 |
| CommonCorpus-Eng | 0.02 | 0.01 | 0.01 | 0.01 | 0.02 | 0.02 | 0.04 | 0.05 | 0.12 | 0.16 |
| C4 | 0.01 | 0.01 | 0.01 | 0.02 | 0.00 | 0.00 | 0.00 | 0.00 | 0.20 | 0.21 |
| DCLM | 0.01 | 0.02 | 0.02 | 0.02 | 0.01 | 0.00 | 0.02 | 0.02 | 0.12 | 0.13 |
| FineWeb | 0.02 | 0.01 | 0.03 | 0.03 | 0.00 | 0.00 | 0.00 | 0.00 | 0.18 | 0.20 |
| HPLT | 0.02 | 0.02 | 0.02 | 0.02 | 0.00 | 0.00 | 0.00 | 0.00 | 0.17 | 0.21 |
| Nemotron-CC-HQ | 0.03 | 0.02 | 0.03 | 0.03 | 0.00 | 0.00 | 0.00 | 0.00 | 0.09 | 0.10 |
| SlimPajama | 0.02 | 0.02 | 0.02 | 0.02 | 0.00 | 0.00 | 0.00 | 0.00 | 0.14 | 0.15 |

Table 10: Benchmark test set sizes (number of examples) for the original benchmarks versus the final decontaminated versions. The 'Decontaminated' column shows the reduced set size after removing all examples with detected 13-gram training data overlap.

| Dataset | Original | Decontaminated |
|---|---|---|
| MBPP | 500 | 331 |
| IFEval | 541 | 429 |
| GSM8K | 1319 | 1235 |
| MBPP+ | 378 | 339 |

## D.4 DECONTAMINATION CASE STUDY

To further alleviate concerns about contamination issues, we performed an experiment where we trained a 1.7B model on a version of MixtureVitae that excludes three dataset shards that contribute 27% of the total contaminated docs— which in particular showed MMLU contamination rates that are high relative to the rest of the dataset, as shown in Table 11. The shards we removed were **Misc-Instruct**, **DART-Math** and **Nemotron Science & Math**. The results are shown in Figure 8 and demonstrate that removing these shards had no effect on MMLU performance.

Table 11: Contamination sources for the **MMLU** benchmark in MixtureVitae, sorted by the number of contaminated documents, high to low.

| Dataset Shard | Contaminated Docs |
|---|---|
| Misc-Instruct | 14,649 |
| DART-Math (Ge et al., 2024) | 11,102 |
| Nemotron Science & Math (Bercovich et al., 2025) | 4,793 |
| MAGACorpus (Xu et al., 2024b) | 241 |
| (All Remaining) | 3,137 |

## D.5 Full Decontamination Experiment

We also performed a stronger decontamination experiment in which we removed *every* document in MixtureVitae that was flagged as contaminated by our 13-gram procedure (Appendix D.2) and retrained a 1.7B model for 300B tokens under the `open-sci-ref` setup. The results—shown in Figure 9—indicate that the fully decontaminated variant performs slightly *better* than the original MixtureVitae model, further addressing concerns that our benchmark results might be inflated by data leakage.

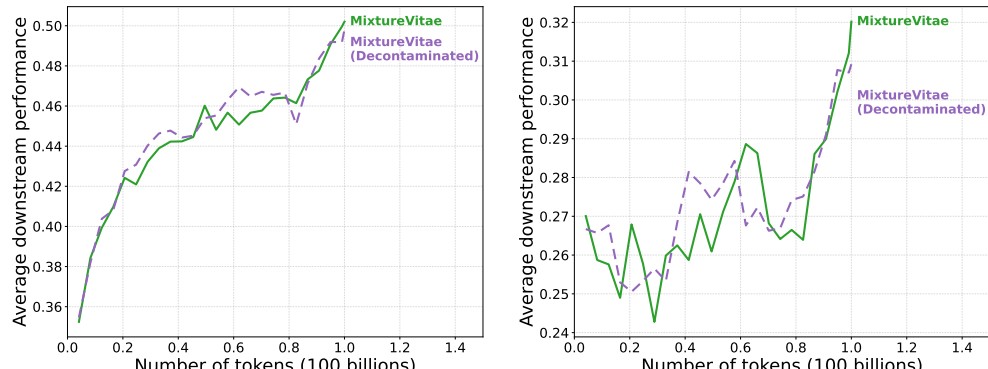

(a) Average accuracy across all tasks (as listed in Ta-ble 5) as a function of number of training steps.

(b) Accuracy on MMLU as a function of number of training steps.

Figure 8: **Validation of 1.7B model performance**. The **MixtureVitae (Decontaminated)** model (purple, dashed), trained with dataset shards responsible for benchmark leakage removed, performs closely to the full **MixtureVitae** (green, solid) model. This confirms our results are not an artifact of test set leakage.

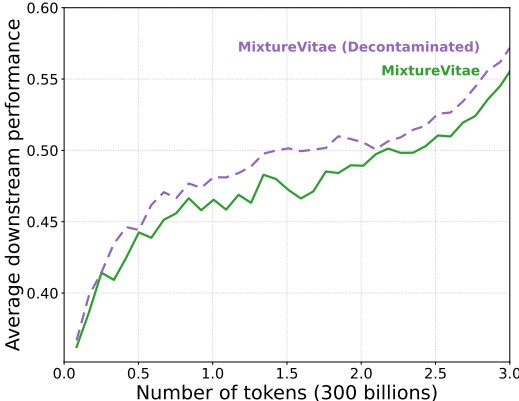

Figure 9: **1.7B model performance on a fully decontaminated dataset**. The model trained on the fully decontaminated **MixtureVitae** corpus (purple, dashed) performs slightly better than the model trained on the full **MixtureVitae** dataset (green, solid), further indicating that benchmark gains are not driven by contaminated examples.

### D.5   DISCUSSION ON DECONTAMINATION METHODOLOGY

Our decontamination pipeline employs the standard 13-gram exact-match procedure (Abdin et al., 2024) to ensure high precision, scalability and comparability with prior baselines. While we acknowledge that exact matching overlooks paraphrased content, we avoided approximate methods (e.g., LSH, embedding-based) due to their tendency to produce false positives on common factual or algorithmic templates (Lee et al., 2022). As noted in Lozhkov et al. (2024a), aggressive removal of semantically similar content risks distorting the training distribution by discarding high-value instructional data.

## E   ABLATION STUDIES

To isolate the impact of primary data components in **MixtureVitae**, we define **Web** and **Instructions** subsets (see Figure 1; **Instructions** encompasses Reasoning & Instruction and Math parts of full mixture) and conduct an ablation study on a 100B-token scale. We train three separate models:

(1) **MixtureVitae (full)**, the complete dataset; (2) **MixtureVitae (w/o Web)**, removing the **Web** component; (3) **MixtureVitae (w/o Instructions)**, removing the **Instructions** component.

The average downstream performance of these models (Figure 10a) shows varying contributions by each component: The **Instructions** data is the most critical driver of performance, as its removal results in the largest, consistent drop of average performance compared to other configurations. Removing **Instructions** particularly leads to severe drop on GSM8k (from $0.47$ to $0.03$) and MBPP, as shown in Figure 10b. Absent the **Instructions** data, the model fails to match the gains of the full mix, underscoring the essential role of instruction-following data in generalization.

Removing the **Web** component (**w/o Web**, blue dashed line) also results in a performance drop below the full dataset, albeit less dramatically. Figure 10b shows a drop from $0.47$ to $0.41$ on GSM8k, far less severe than the drop close to 0 for **w/o Instructions** and only slight changes on code evals. The comparison of ablation effects again highlights the crucial role of **Instructions** data.

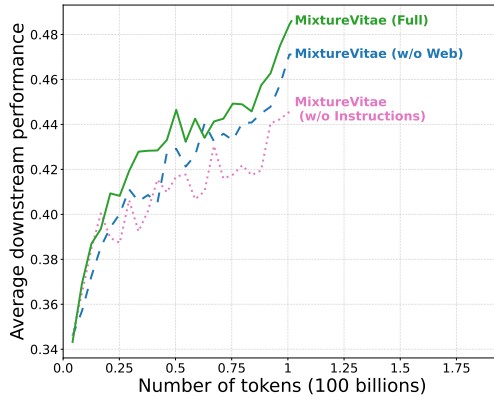

(b) A performance breakdown on math, coding and instruction following tasks for the ablated dataset variants. Best results are in bold.

| Training Dataset | IF-Eval | GSM8K | MBPP | Average |
|---|---|---|---|---|
| **MixtureVitae** | 0.14 | **0.47** | **0.34** | **0.25** |
| **MixtureVitae** (w/o Web) | 0.18 | 0.41 | 0.33 | **0.25** |
| **MixtureVitae** (w/o Instructions) | **0.19** | 0.03 | 0.14 | 0.14 |

(a) Ablation on full **MixtureVitae** against two versions, each without a main component. Average performance on 10 downstream tasks.

Figure 10: An ablation study on components of the **MixtureVitae** dataset. Figure 10a shows performance average on 10 downstream evals during training, while Figure 10b shows scores on further math, code and instruction benchmarks. The evaluation setup is given in Table 6.

## F  Synthetic Math Data Generation

The synthetic math dataset was programmatically generated to produce a diverse range of mathematical problems and their solutions. The generation process covers a wide array of mathematical domains, including fundamental arithmetic operations, multi-term fractional expressions, and the step-by-step solution of algebraic linear equations. A key component of the dataset consists of word problems, where numerical challenges are embedded in narrative scenarios.

A significant feature of this generation pipeline is the creation of detailed, step-by-step solutions formatted as a chain-of-thought. For many problem categories, the scripts produce a human-readable explanation of the entire solution process. This is achieved by using a variety of randomized natural language templates to describe each logical step, such as carrying a digit in addition or isolating a variable in an equation.

Following the initial generation, a final post-processing step is applied to format the dataset for model training. This stage programmatically identifies data entries containing human-like, explanatory text by searching for common instructional words. For these selected entries, a descriptive header (e.g., "Here are examples of addition, division exercises") is dynamically generated. The content and phrasing of this header are randomized and based on the mathematical operations present within the text, adding significant linguistic diversity.

For example, in the generated math problem below, a model may be able to generalize to new numbers, but if the problem were to add three students instead of two, the model may not be robust enough to generalize. We leave this analysis for future work.

```
The age difference between Sarah and Asaf's age is half the
total number of pencils Sarah has. The sum of their ages is
132, and Sarah is 27 years old. If Asaf has 60 more pencils than
Sarah, calculate the total number of pencils they have together.
Solution: If the sum of their ages is 132, and Sarah is 27 years
old, Asaf is 105 years old.
The age difference between Sarah
and Asaf's age is 105-27 = 78.
Since the age difference between Sarah and Asaf's age is half
the total number of pencils Sarah has, Sarah has 2*78 = 156 pencils.
If Asaf has 60 more pencils than Sarah, Asaf has 156+60= 216 pencils.
Together, they have 156 + 216 = 372 pencils.
```

# G   OUR POSITION ON USING GOVERNMENTAL AND OTHER WORKS UNDER FAIR USE AND RELATED ETHICAL AND LEGAL BASIS

To contextualize our licensing tiers and clarify the rationale behind including certain higher-risk but legally supportable sources, we outline here the ethical and legal considerations underlying MixtureVitae's construction. Our goal is not to offer legal advice or definitive interpretations of copyright law, but rather to articulate the principles—fair use, permissive upstream licensing, government-works doctrine, and the EU text-and-data-mining (TDM) (The European Parliament and the Council of the European Union, 2019; Margoni & Kretschmer, 2022) exception—that inform our "permissive-first, risk-mitigated" design philosophy. We provide this discussion so downstream users can understand how specific dataset subsets were evaluated and what residual risks remain despite our filtering and provenance-tracking efforts.

## G.1   FAIR USE OF GOVERNMENT WORKS

In order to increase the diversity of our dataset, we included ≈11B tokens of governmental website data from US federal, US non-federal, and non-US government sources. While works created by the US federal government are generally not copyrightable, other governmental website content may neither be expressly in the public domain nor explicitly licensed. For those sources, we rely on fair use principles (United States Congress, 1976 (usc, 1976) ; Lemley & Casey, 2017 (Lemley & Casey, 2021)) and the EU text and data mining exceptions (The European Parliament and the Council of the European Union, 2019; Margoni, 2019), which together mitigate the risk associated with using this subset.

Our ethical and legal reasoning for using this government web content—sourced from Common Crawl–related datasets (Common Crawl Foundation, 2025) that respect *robots.txt* opt-out—is as follows:

- **Public Purpose Alignment**: The content created by governments is normally meant to be shared with the public, and by using the data for training we are assisting this purpose.
- **Purpose of Use**: From a legal perspective, the government works are being redistributed as part of an open source, no-fee dataset, used to create models are less likely to violate copyright. This purpose is clearly not to compete with the government's own usage.
- **Effect on Potential Market**: Our use of government website content is unlikely to affect any potential market for that content, as governments typically do not exploit these materials commercially in ways that would compete with our dataset or downstream models. This factor favors a finding of fair use.
- **Nature of the Content**: The nature of the content is mostly public announcements, content of public interest, governmental functions or the like. Again, we believe there is strong public policy interest for fair use of this type of information.

- **Amount Used**: While we use all or almost all of the content of the government website, the amount of usage is not determinative of fair-use or not fair-use.

- **Federal vs. Non-Federal Works**: Lastly, US works created by the federal governments are generally not copyrightable. However, we recognize that this is not the case for other foreign governmental works, or non-federal works.

For these reasons, we believe using government website data presents relatively lower copyright risk. To further minimize risk—for example, the potential inclusion of third-party copyrighted works embedded in government web pages—we apply keyword filters such as "All Rights Reserved" and "Copyright ©" to exclude pages that contain such terms.

Recent court cases, as of the writing of this paper, include:

- **Bartz v. Anthropic PBC**: district court ruling that use of purchased copies of books for AI training is fair use.

- **Kadrey v. Meta Platforms, Inc.,**: district court ruling that training on authors' books was transformative fair use.

These developments lend some support to the argument that AI training on web-text data—including our relatively small, public-facing government subset—can fall within fair use, though the case law is still evolving.

### G.2 OTHER TIER-2 DATA WITH OPAQUE OR MIXED PROVENANCE

Similarly, our dataset includes data whose provenance is not entirely transparent even though the license on the upstream dataset appears permissive, such as `The Stack V1` and other Tier-2 sources identified in Appendix J. In the case of `The Stack V1`, a line-by-line audit to remove copyrighted content has not been performed, and therefore some risk remains in its usage. Nonetheless, we rely on fair use to justify this usage because the data are used to train models, rather than to provide a substitutive or competing software product. For a more detailed discussion of `The Stack V1`, see Section H.

For other Tier-2 data, some upstream generator models impose conditions on downstream use—such as the Llama license, which requires model users to adhere to certain limitations. We do not believe we are bound by terms that were not contractually passed through to us by our direct licensor, although this issue is subject to debate. We therefore classify this small portion ($\approx$4%) of the dataset as **Tier 2(b)**.

There are additional Tier-2 data where the provenance is partially opaque. For example, a small portion of our P3 dataset, when converted into a few-shot format, may pose higher risk than other Tier-1 data. While the ultimate source datasets that constitute P3 are well-known academic benchmarks, some of those component datasets do not provide explicit licenses. Nonetheless, we consider the resulting few-shot datasets to be highly transformative and unlikely to compete with the underlying works: they are mixed and reformatted multiple times for the specific purpose of training classification and few-shot models, rather than, for example, serving as standalone product reviews. We classify these higher-risk works as **Tier 2(b)** and include them in our dataset with that caveat.

### G.3 RELIANCE ON EU TEXT AND DATA MINING EXCEPTIONS

We also rely, to some extent, on the EU text and data mining exception (Margoni & Kretschmer, 2022) for our inclusion of web-crawled data. This regime is complementary to US fair-use doctrine, and we mention it here for completeness. In particular, we depend on Common Crawl's practice of respecting *robots.txt* at the time of crawling. We do not believe retroactive recrawling is legally necessary to determine whether a work was subsequently opted out, but we nonetheless commend efforts towards doing so, such as Apertus (Hernández-Cano et al., 2025).

### G.4 Residual Copyright and Trademark Risks

The copyright risks in machine learning are complex. For example, copyrighted materials may appear as limited fair-use quotations in Wikipedia articles [2]. A model trained on such materials in the aggregate could, in principle, generate more substantial and potentially infringing text than the short quotations present in the dataset. Future work should address this risk, including (i) copyright evaluation audits of datasets, and (ii) model-level mitigations that encourage limited direct quotation and discourage reproduction of substantial protected passages.

As with other large, permissively licensed datasets, additional legal risks remain, including trademark risks. For instance, while training on a Wikipedia article about "Spiderman" may be relatively low risk (given its CC-BY-SA license and the educational, summarizing nature of the article), a model that subsequently generates new stories featuring the character name "Spiderman"—even if the plots themselves are not derived from existing human-created stories—may still implicate trademark rights. Addressing those issues thoroughly is beyond the scope of this work and is left for future research.

We do not and cannot guarantee that, even with rigorous provenance tracking and standard filtering, the dataset is free of legal risk. Nothing in this section constitutes legal advice. We recommend that anyone who uses our datasets consult their own legal counsel in their jurisdiction before deploying models trained on this data in commercial settings.

## H Provenance and Rationale for The Stack v1 (OpenRAIL-M and terms of use)

Our inclusion of 53.2B tokens sourced from **The Stack v1** (Kocetkov et al., 2022a), which we categorize by its governing dataset card terms of use and which subsequent model uses the **OpenRAIL-M** license, warrants this specific note on provenance. The data was included based on the following rationale:

- **Source and Filtering Methodology:** The dataset originates from a large-scale scrape of GitHub. The BigCode project curated this data by applying a filter to include only those repositories that contained a clear permissive license file (e.g., MIT, Apache 2.0, BSD) at the root level.
- **Acknowledged Heuristic:** This repository-level filtering is a *heuristic* and not a file-level guarantee. As acknowledged by the dataset's creators, this process cannot perfectly resolve complex cases of multi-licensing within a single repository, such as the inclusion of non-permissively licensed vendor libraries or mixed-license assets alongside permissively-licensed code.
- **Inclusion Justification:** Despite this caveat, The Stack v1 represents the largest-available public corpus curated with the *explicit goal* of permissive filtering. Excluding it would make training a high-performance, open, and risk-mitigated code model nearly impossible. Its "best-effort" permissive curation philosophy directly aligns with our dataset's core principle of risk-mitigation.

Thus we include it in our dataset with the classification of Tier-2, as defined in Section 2.1.4.

## I Data Filtering Reasoning and Protocol

To promote transparency, we describe our protocol for defining and checking the lists and content of the pseudo-crawled portion of MixtureVitae.

### I.1 Governmental and NGO Domain Patterns

The following list of URL patterns was used to filter for governmental, non-governmental, and international organization websites from the web datasets. We gathered the list by examining public

---

[2]https://en.wikipedia.org/wiki/Wikipedia:Quotations#Copyrighted_material_and_fair_use

records, Wikipedia lists, and the like. The list is not as simple as **gov.** because international governments use different TLDs. Moreover, some spam websites masquerades as **.gov** websites. Two of the authors examined each domain either online or through the *Internet Archives' Way Back Machine* to confirm they belonged to a government website. After performing a pseudo-crawl on FineFineWeb, Nemotron-CC and MagaCorpus, the authors manually audited the data for quality, and filtered out spam websites with similar website names, which were added to blocklists.

The **.gov, .gov/, and .mil/** websites are US Federal governmental works. To the extent we could, we filtered any sites that had keywords indicating reservations of rights. We believe this lowers the risk of inadvertent third party copyrighted works appearing on US Federal works, and is in the spirit of the EU text data mining opt-out conventions. We also note that the ultimate source of these websites is from Common Crawl which already also respects the *robots.txt* opt-out.

- `gov` (as a suffix)
- `gov/`
- `mil/`

All other websites in this category are specifically international governments or NGOs.

```
vlada.mk, vlada.cz, kormany.hu, regeringen.*, rijksoverheid.nl, government.nl,
bund.de, bundesregierung.de, government.ru, gc.ca, admin.ch, www.gob.cl/,
www.gob.ec/,    guatemala.gob.gt/,    presidencia.gob.hn/,    www.gob.mx/,
presidencia.gob.pa/, www.gob.pe/, gob.es/, argentina.gob.ar/, tanzania.go.tz/,
indonesia.go.id/, go.kr/, go.jp/, thailand.go.th/, europa.eu/, un/, int/, govt.,
www.gub.uy, gov., gouv.
```

## I.2 CURATED PERMISSIVE DOMAIN LIST

The following list of approximately 50 domains was curated based on their known public domain or CC-BY-SA* license status or a permissive status. The websites were chosen for their diversity of content. Two of the authors—one of which has a legal background—examined the websites' terms of use, or relevant sections online or on the Way Back Machine to confirm licensing and permission status. After performing a pseudo-crawl on FineFineWeb, Nemotron-CC and MagaCorpus, the authors manually reviewed the data for quality, and filtered out spam websites with similar website names as the below. These spam sites were added to blocklists.

```
free.law,    europeana.eu,    publicdomainreview.org,    wisdomcommons.org,
intratext.com, mediawiki.org, wikimedia.org, wikidata.org, wikipedia.org,
wikisource.org,    wikifunctions.org,    wikiquote.org,    wikinews.org,
wikivoyage.org, wiktionary.org, wikibooks.org, ,courtlistener.com/³, case.law,
pressbooks.oer.hawaii.edu,    huggingface.co/docs,    opencourselibrary.org,
medbiq.org,    doabooks.org,    bccampus.ca,    open.umn.edu/opentextbooks,
www.gutenberg.org, mozilla.org, www.eclipse.org, apache.org, python.org,
pytorch.org, numpy.org, scipy.org, opencv.org, scikit-learn.org, pydata.org,
matplotlib.org, palletsprojects.com, sqlalchemy.org, pypi.org, sympy.org,
nltk.org, scrapy.org, owasp.org, creativecommons.org, wikia.com, foodista.com,
fandom.com, attack.mitre.org
```

The vast majority of these sites are CC-BY licensed. However, there are some that have other open licenses as shown in Table 12.

## J SYNTHETIC DATA SOURCE PROVENANCE

To ensure full transparency regarding the "permissive-first" nature of **MixtureVitae**, we provide a detailed provenance audit of our synthetic data components in Table 13, including classification to tiers as defined in Section 2.1.4.

To validate the robustness of our permissive-first strategy, we further analyze the contribution of synthetic components categorized as **Tier 2(b)**. This small part of **MixtureVitae** is comprised of subsets which are permissively licensed (e.g., Apache 2.0) but are derived from generator models

---

³For `courtlistener.com`, the terms of use says it is CC-BY-ND, but the underlying court cases are public domain, and the content from this website is merely 176KB and is de minimus.

Table 12: Software Licenses and Associated Websites

| License | Websites |
|---|---|
| BSD 3-Clause | scipy.org, sympy.org, matplotlib.org, scrapy.org, scikit-learn.org, pydata.org, pytorch.org, palletsprojects.com |
| Mozilla Public License | mozilla.org |
| Python Software Foundation License 2.0 | python.org |
| Apache 2.0 | apache.org, nltk.org, opencv.org |
| MIT License | sqlalchemy.org |
| Eclipse Public License | www.eclipse.org |
| MedBiquitous Standards Public License | medbiq.org |

with restrictive community licenses (such as Llama-3) or seed data with partially opaque origins. As illustrated in Figure 11, removing these Tier 2(b) components yields a training trajectory indistinguishable from the full **MixtureVitae** baseline. This result confirms that our model's strong performance is driven by its core, fully verifiable permissive sources, ensuring that users with strict compliance requirements can safely exclude Tier 2(b) data without compromising downstream quality.

See Section G for a further discussion on our justification for including Tier 2 and in particular Tier 2(b) data.

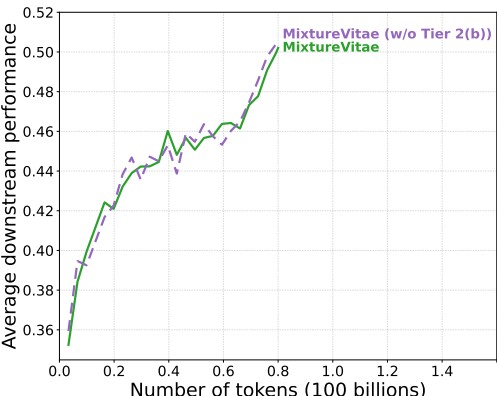

Figure 11: **Ablation of Tier 2(b) components.** We compare the training trajectory of the full **MixtureVitae** dataset (green) against a version excluding Tier 2(b) (purple dashed). Tier 2(b) consists of synthetic data derived from non-permissive generators (e.g., Llama-3) or seeds with opaque provenance. The nearly identical performance curves demonstrate that users requiring strict permissive compliance can exclude these components with negligible impact on downstream model quality.

## K  SCALING OUTLOOK AND FUTURE DIRECTIONS

While **MixtureVitae** currently comprises 422 billion tokens—a scale smaller than frontier runs which often exceed 10 trillion tokens—our primary objective in this work was to establish a proof-of-concept for data efficiency and strong downstream performance within a strict permissive-first, risk-mitigated licensing framework. We identify several concrete avenues to scale this approach to the multi-trillion token regime required for larger foundation models:

**Subset Upsampling.** Standard industry recipes for large-scale training often heavily upsample high-quality data. For instance, Llama 3 (Meta, 2024) employs upsampling factors of 4–10× for its highest-quality subsets to reach its training budget. In contrast, the current iteration of **MixtureVitae** does not assign aggressive upsampling factors to individual shards. Applying standard upsampling

Table 13: Detailed provenance of synthetic data sources in **MixtureVitae**.

| Dataset Name | Dataset License | Model | Seed Data Provenance | Token Count(B) | Notes |
|---|---|---|---|---|---|
| **Tier 1: Fully Permissive (≈ 161B Tokens)** | | | | | |
| GlaiveAI Reasoning | Apache 2.0 | Permissive | N/A | 38.366 | Fully synthetic |
| Nemotron (Science & Math) | CC-BY-4.0 | Permissive | Permissive (StackOverflow, WildChat) | 22.310 | Science & Math subset of Llama-Nemotron-Post-Training-Dataset |
| Ling-Coder/SyntheticQA | Apache 2.0 | Permissive | N/A | 19.852 | |
| Open Thoughts | Apache 2.0 | Permissive | Permissive (OpenMath-2-Math, CodeGolf, OpenCode, etc) | 18.786 | Excludes Organic Chemistry subset |
| EuroPat | Public Domain | Permissive | Permissive | 11.586 | Synthetic image captions created from patents |
| P3 (Permissive Subset) | Apache 2.0 | N/A | Permissive (ARC, PIQA, BoolQ, etc) | 10.130 | |
| Nemotron-CC | Common Crawl ToS | Permissive | Permissive (Common Crawl) | 6.230 | Using a Permissive-only subset |
| YouTube | CC-BY-4.0 | Permissive | Permissive (VALID, CommonCorpus) | 7.386 | Derived from CC-BY licensed YouTube content |
| Prism-Math | CC-BY-4.0 | Permissive | Permissive (NuminaMath-1.5) | 5.682 | |
| DeepMind Math | Apache 2.0 | N/A | Permissive (Procedurally Generated) | 4.232 | |
| Misc. Instruct. / NVidia OpenMathInstruct-1 | NVIDIA license | Permissive | Permissive (GSM8K, MATH) | 2.440 | |
| Websights | CC-BY-4.0 | Permissive | N/A | 1.018 | Fully synthetic |
| Misc Instruct. / MetaMathQA-R1 (responses) | MIT | Permissive | Permissive (GSM8K, MATH) | 0.672 | |
| Math Word Problems | Apache 2.0 | N/A | Permissive | 0.456 | Procedurally Generated |
| Ling-Coder/DPO | Apache 2.0 | Permissive | Unknown (Common-Crawl) | 0.398 | |
| Misc. Instruct. / OpenR1-Math-220k | Apache 2.0 | Permissive | Permissive (NuminaMath-1.5) | 0.320 | |
| Misc. Instruct. / NVIDIA SFT Datablend | CC-BY-4.0 | Permissive | Permissive (MNLI, COPA, PIQA, etc) | 0.286 | |
| Misc. Instruct. / OpenThoughts-114k-Code (decontaminated) | Apache 2.0 | Permissive | Permissive (TACO, Apps , CodeContests, etc) | 0.150 | |
| Misc. Instruct. / Synthetic Code Generations | Apache 2.0 | Permissive | N/A | 0.104 | Fully synthetic |
| Misc. Instruct. / PrimeIntellect StackExchange QnA | Apache 2.0 | N/A | N/A | 0.076 | |
| Misc. Instruct. / PrimeIntellect Real World SWE Problems | Apache 2.0 | Permissive | Permissive (CommitPack) | 0.006 | |
| Misc. Instruct. / PrimeIntellect Synthetic Code Understanding | Apache 2.0 | Permissive | N/A | 0.004 | Fully synthetic |
| Misc. Instruct. / GSM8K (train) | MIT | Permissive | N/A | 0.004 | Fully synthetic |
| **Tier 2(a): Permissive with Upstream Opacity (≈ 35B Tokens)** | | | | | |
| OS-Q2 (OpenScience) | CC-BY-4.0 | Permissive | N/A | 17.366 | Fully synthetic |
| Ring-lite SFT Data | Apache 2.0 | Permissive | Permissive (CodeContest, APPS, TACO, etc) | 14.968 | |
| PyEdu Reasoning | Stack V1, ODC-BY | Permissive | Permissive (The Stack V1) | 3.138 | |
| Misc. Instruct. / Magpie-Phi3-Pro-1M-v0.1 | N/A | Permissive | N/A | 0.386 | Fully synthetic |
| MegaWika | CC-By-SA/4.0 | Permissive | Permissive (Wikipedia) | 0.356 | |
| Misc. Instruct. / Magpie-Qwen2.5-Coder-Pro-300K-v0.1 | N/A | Permissive | N/A | 0.120 | Fully synthetic |
| Misc. Instruct. / NovaSky-AI Sky-T1 | Apache 2.0 | Permissive | Permissive (AIME, MATH, etc) | 0.080 | |
| Misc. Instruct. / BigCode Self-OSS-Instruct | Stack v1 | Permissive | Permissive | 0.018 | |
| Misc. Instruct. / UltraFeedback | MIT | Permissive | Permissive (UltraChat, TruthfulQA, etc) | 0.014 | |
| Misc. Instruct. / CaseHOLD (Phi4 Reasoning Traces) | N/A | Permissive | Permissive (CaseHOLD) | 0.002 | Case law is public domain |
| **Tier 2(b): Restricted, Mixed or Opaque Provenance (≈ 17B Tokens)** | | | | | |
| Ling-Coder/SFT | Apache 2.0 | Permissive | Partially Unknown (Github, CommonCrawl, The Stack, etc) | 7.668 | Unknown provenance of CommonCrawl subset |
| P3 (Commercial Subset) | Apache 2.0 | N/A | Ambiguous-license (Amazon, Rotten Tomatoes, IMDb, etc) | 5.730 | User-generated content on commercial platforms with no verifiable licenses |
| Misc. Instruct. / OpenMathInstruct-2 | CC-By-4.0 | Restricted | Permissive (GSM8K and MATH) | 3.202 | Llama-generated responses |
| Misc. Instruct. / OpenManus-RL | Apache 2.0 | Mixed | Permissive (AgentTraj-L, Agent-FLAN, etc) | 0.024 | GPT-4 used for generating traces in the AgentTraj-L subset |
| **Tier 3: Civic / Governmental Works (≈ 15B Tokens)** | | | | | |
| Nemotron-CC (Gov. Portion) | Common Crawl ToS | Permissive | Permissive (Common Crawl) | 8.531 | Using a Permissive-only subset |
| MagaCorpus | ODC-By | Permissive | Permissive (FinewebEdu-dedup) | 6.596 | Using only permissive subsets |

techniques to our highest-value subsets (such as curated reasoning) would immediately scale their contribution to the total token count.

**Multilingual Expansion.** The current release of **MixtureVitae** is primarily English-centric. Expanding the sourcing strategy to include multilingual data represents an order-of-magnitude opportunity for scaling. This can be achieved through two primary methods: (1) identifying and allowing international permissively licensed sources, and (2) using machine translation to expand the existing data in **MixtureVitae**.

**Synthetic Expansion.** Our Math and Reasoning synthetic subsets are generated procedurally or via LLMs. This generation process is horizontally scalable. By increasing the compute budget for generation, these high-density subsets can be expanded significantly without incurring the legal risks associated with scraping organic web data.

**Web Data Rephrasing.** Recent work has demonstrated the utility of rephrasing web data to improve quality and standardize style (Maini et al., 2024). Applying a similar rephrasing pipeline on top of the **MixtureVitae** web data processing pipeline can further expand the corpus volume while maintaining the strict safety and licensing standards defined in our framework.

