# OpenReview forum: "MixtureVitae: Open Web-Scale Pretraining Dataset With High Quality Instruction and Reasoning Data Built from Permissive Text Sources"
_ICLR.cc/2026/Conference — ICLR 2026 Conference Desk Rejected Submission_

### Official Review · Reviewer_otHz · 2025-10-29

**Soundness:** 3
**Presentation:** 4
**Contribution:** 3
**Rating:** 8
**Confidence:** 3

**Summary:**

This paper curates a large-scale open and permissive pretraining corpus. In comparison to its permissive counterparts, Mixture Vitae has a special focus on reasoning/math/code- specific data and synthetic data, giving it superior performance in corresponding downstream tasks. The authors have also conducted rigorous experiments in an open-sci-ref manner, which shows that Mixture Vitae serves as a strong permissive baseline and sheds insights on which part of the data brings in the biggest performance improvement. Overall, the paper makes an important contribution to the open-source community.

**Strengths:**

-The authors make a timely and commendable contribution toward creating legally safer and more reproducible pretraining data amid increasing copyright litigation risks.

- The paper shows that a permissive-first data approach can produce competitive models, with the proposed dataset outperforming all other permissive alternatives.

**Weaknesses:**

- It seems that a major strength of the MixtureVitae dataset is to minimize copyright risk; however, there are no demonstrative experiments to support this. For example, it would be interesting to see if memorisation over New York Times articles instead gets reduced training with MixtureVitae. Similar verbatim experiments like in [1,2] can further strengthen this paper.

- As a pretraining dataset, its size is relatively small compared to the 8TB used in [3].

**Questions:**

- Regarding the web portion of your dataset, how did you check the licenses of the crawled dataset? In a recent paper by Fan et al [2], the authors did a retroactive robots.txt filtering to make sure web-scale data is compliant. Have similar efforts been done here?

- I think it is pretty impressive how much improvement one can get by curating more code/math/reasoning data. However, is it necessarily the goal of pretraining to arrive at a reasoning-capable base model? A lot of models only enhance such capabilities in post training, and it seems to me that they just used such data in their post training phases. Thus it is unfair to compare the performances on GSM-8k, MBPP, IF-eval such benchmarks.



[1] Exploring Memorization and Copyright Violation in Frontier LLMs: A Study of the New York Times v. OpenAI 2023 Lawsuit

[2] Can Performant LLMs Be Ethical? Quantifying the Impact of Web Crawling Opt-Outs

[3] The Common Pile v0.1: An 8TB Dataset of Public Domain and Openly Licensed Text

---

> ### Author Response · Authors · 2025-11-20
>
> We thank the reviewer for their feedback and for recognizing our work as a "timely and commendable contribution" that establishes a "strong permissive baseline". We address the specific concerns below.
>
> ### Dataset Size and Scalability
>
> We acknowledge the observation regarding the size of our dataset (211B tokens). As discussed in our general response, our aim in this work is not to match current 10T‑token frontier runs, but to demonstrate data‑efficient training under a permissive‑first, risk‑mitigated licensing framework that is competitive with non-permissive datasets. We outline several paths on how to scale up the presented dataset composition (upsampling, multilingual expansion, synthetic growth, translation) for future versions. In regard to the Common Pile: while it does contain ~8TB of data, it is a data pool, not a training set. Comma-0.1, a permissive training set derived from Common Pile, contains ~600B unique tokens, with subsets repeated to create ~2T tokens. Since our comparative evaluation relies on a fixed 300B token budget, MixtureVitae’s current scale is well-suited for establishing a valid performance ranking against open reference datasets.
> ### How did we check the licenses of the crawled dataset?
> Our approach to license verification differs from the negative filtering typically applied to broad web scrapes (e.g., Fan et al.). We employ a permissive-first, upfront sourcing strategy:
> * **Positive Inclusion**: Rather than ingesting broad web dumps and applying retroactive negative filtering, we positively select safe sources derived from Common Crawl. As detailed in Section 2.2.1, we performed a "pseudo-crawl" based on explicit allowlists, manually verifying domains prior to inclusion, ensuring the license status is auditable at the source level.
> * **Keyword-Based Verification**: For broader expansion, we further enforced strict positive inclusion criteria. Where the URL provenance was unknown, documents were required to contain permissive license keywords to be eligible. We further applied negative filters for restrictive terms.
>
> ### NYT and Memorization
> Our dataset composition makes memorization of restricted news sources unlikely. Over 90% of MixtureVitae consists of domains where commercial news is naturally absent (code, math, etc). The web component (~8%) was subject to a positive inclusion strategy rather than passive ingestion, while being sourced mostly from government websites. Therefore, we expect the presence of paywalled or opted-out publishers to be very low, and NYT-style memorization tests are therefore less diagnostic for MixtureVitae than for broad web scrapes.
>
> **Action**: We will expand Section 2.2 to explicitly contrast our "Positive Inclusion" strategy with standard "Negative Filtering".
>
> ### Why include instructions in pre-training? Is it fair to compare to base models only despite training on instruct-like data?
>
> * **Motivation for Reasoning/Instruction During Pretraining**: We acknowledge the traditional view that reasoning data is often reserved for post-training. However, our approach aligns with recent work (e.g. Phi-4) demonstrating that front‑loading reasoning and instruction data into pre‑training can yield substantially stronger small models than relying on post‑training alone.
> * **Methodological Validity**:  Our intent is not to claim that our base model is directly comparable to a “web scrape” baseline that has never seen instruction‑like examples. Rather, the object of study in this paper is how far one can go by changing only the pre‑training corpus, holding architectures and training budgets fixed and adhering to a “permissive-first, risk-mitigated” framework. Evaluating the resulting model on GSM8K and MBPP is therefore a way to measure the effect of this corpus design on math and code capabilities.
> * **Fairness of Comparison and Relation to Instruction‑Tuned Models**: To address the concern that we are comparing a “reasoning‑heavy” pre‑trained model against bare base models, we also include a comparison to SmolLM2‑1.7B‑Instruct, a strong instruction‑tuned model trained on ≈11T tokens. Despite SmolLM2‑Instruct utilizing a massively larger compute budget (~11T) and a multi‑stage post‑training pipeline, our single‑stage MixtureVitae 1.7B model (300B tokens) matches or exceeds it:
> | Model | GSM8K | MBPP |
> | :--- | :---: | :---: |
> | **SmolLM2-1.7B-Instruct** (11T) | 37% | 37% |
> | **MixtureVitae-1.7B-Base** (300B) | **53%** | **38%** |
>
> In summary, our results indicate that high‑quality, risk‑mitigated data is a token‑efficient alternative to broad web scrapes for building reasoning‑capable small models.
>
> **Action**: we will clarify in the revised manuscript that our “base” model is best described as pre‑trained with a reasoning/instruction‑heavy, permissive‑first mixture, and we will more clearly distinguish this from web‑only baselines in the text and captions.
>
> Once again, we thank you for your thorough review and constructive feedback.

---

> > ### Comment · Reviewer_otHz · 2025-11-27
> >
> > Thank you for the clarification. I have no additional concerns. Please incorporate these points into the final version of the script.

---

### Official Review · Reviewer_1Bso · 2025-10-31

**Soundness:** 4
**Presentation:** 3
**Contribution:** 3
**Rating:** 8
**Confidence:** 4

**Summary:**

This paper introduces MixtureVitae, a pre-training corpus of 211B permissively licensed data, and thus minimize the risk of copyright. To enrich the diversity and quality of MixtureVitae, they aggregate existing permissive high-quality data resources, synthetic data, instruction-following data, and reasoning data. They conduct pre-training experiments using MixtureVitae and baseline datasets on 4 sizes of LLMs (from 130M to 1.7B), finding that MixtureVitae leads to better pre-training outcome than other permissive pre-training corpus. The performance is even competitive to high-quality, non-permissive baselines.

**Strengths:**

- Originality and significance: the paper's creativity can be reflected from the data curation pipeline. It strategically combines data from various sources and published datasets, including web data, synthetic data, instruction-following, and reasoning data. More importantly, such aggregation is under a license-permissive manner to minimize the potential legal risk. It addresses a critical problem in the field: how to train high-quality open LLMs on permissive data only. To my knowledge, only concurrent work Apertus tries to address this problem despite its importance.

- Quality: The paper conduct comprehensive evaluation and comparison across difference LLM sizes, training data scale, and evaluation benchmarks. It also involves robustness check through decontamination and ablation. The permission filtering is particularly novel compared to related work.

- Clarity: The data processing pipeline -- the major contribution of this pre-training corpus paper -- is well documented and clearly understandable.

**Weaknesses:**

I'm not aware of major weaknesses hurting the soundness this paper. One minor weakness:

- Lack of discussion on the potential scalability of the data collecting pipeline: current LLMs need ~10T level tokens to achieve SOTA results. However, MixtureVitae only has 211B tokens. The size is fair considering its permissive feature, but would be good to discuss whether the quantity can further grow using the proposed data curation pipeline. The limited datasize also restrict the experiments to LLMs of relatively small scales.

**Questions:**

- Differences between MixtureVitae and Apertus: It is fair enough not including Apertus dataset as a baseline, given that it is a concurrent work. But the related work discussion would be much more thorough, if the paper can discuss more about how the data collection of Apertus and MixtureVitae differs, and how the advantages of both can be combined together.

**Details Of Ethics Concerns:**

I am not a legal expert in copyright and licensing, which is a main claim of the paper. So it would be good if someone else can check it.

---

> ### Author Response · Authors · 2025-11-20
>
> We thank the reviewer for their constructive feedback and for introducing the comparison to the Apertus project. We appreciate the opportunity to clarify our positioning regarding dataset scale and to contextualize our contribution within the landscape of open-source data initiatives. We address the specific concerns below.
>
> ### Dataset Scale
>
> We acknowledge the reviewer’s observation regarding the size of our dataset (211B tokens) compared to large-scale web crawls. We addressed this concern in our general response to all reviewers. It is also worth noting that our 211B count represents unique tokens; following standard industry practices (e.g., Llama 3) of upsampling high-quality subsets would immediately multiply our effective training volume.
>
> **Action**: We will incorporate the discussion on dataset scalability into the Discussion section to address the token-count concern.
>
> ### Comparison with Apertus
>
> We thank the reviewer for highlighting Apertus. We agree that it is a relevant and important parallel. We wish to clarify that while both projects value openness, they represent distinct, yet complementary, design philosophies:
> * **Scale vs. Efficiency**: Apertus optimizes for breadth and scale, processing 15T tokens across 1800+ languages to maximize knowledge capacity. In contrast, MixtureVitae (211B tokens) focuses on data efficiency and English‑centric reasoning density. Our results show that a Phi‑style, reasoning‑heavy mixture of high‑quality permissive sources can achieve strong performance on MMLU, GSM8K, and MBPP with roughly 1.5% of the pretraining token budget of a 10T–15T token dataset.
> * **Compliance Strategy:** Both projects aim for legally safer data, but they implement this in different ways. Apertus builds on large web‑scale datasets (e.g., FineWeb‑2, DCLM‑Edu) and applies retroactive robots.txt and limited license filtering to exclude copyrighted content.  MixtureVitae instead uses a positive inclusion (allowlist) strategy: we curate sources whose licensing and use case are known to be permissive (government works, The Stack, permissive benchmarks, vetted synthetic datasets).
> * **Openness and Packaging of Training Data:** Apertus makes a major contribution by releasing training data components and reconstruction scripts. MixtureVitae complements this by releasing a single ready-to-use Apache‑2.0 pretraining mixture. This makes it straightforward for downstream users to reproduce our experiments.
> * **Synergy and Future Directions:** We view MixtureVitae and Apertus as synergistic rather than competing. Apertus shows that large‑scale, multilingual, license‑filtered web data can support strong 8B/70B models at 15T tokens, while MixtureVitae explores how a reasoning‑heavy, permissive‑first mixture at 211B tokens can drive strong performance. Combining these perspectives could yield even stronger compliant models at larger scales.
>
> We hope this clarifies the distinction between our approach, focused on data efficiency and strong reasoning, and broad-scale initiatives like Apertus. We believe both perspectives are valuable for the open-source ecosystem and thank the reviewer again for helping us refine this positioning.
>
> **Action**: we will clarify the differentiation between MixtureVitae and Apertus in the Related Work section.
>
> Once again, we thank the reviewer for their helpful feedback and we hope that we had addressed all concerns.

---

### Official Review · Reviewer_NdNQ · 2025-11-01

**Soundness:** 3
**Presentation:** 4
**Contribution:** 4
**Rating:** 8
**Confidence:** 4

**Summary:**

This paper introduces MixtureVitae, a 211.1 B token permissive-first, risk-mitigated pretraining corpus that integrates explicitly licensed sources (CC-BY, Apache, public domain), curated permissive code, civic/governmental content, and reasoning-oriented synthetic data with documented provenance. Using a controlled open-sci-ref training recipe, the authors show that MixtureVitae (i) outperforms existing permissive datasets and (ii) approaches parity with non-permissive corpora such as Nemotron-CC-HQ and DCLM, particularly on MMLU, GSM8K, and MBPP. The dataset composition, license tiers, filtering pipeline, and decontamination analyses are fully documented. The main contribution is an empirically validated demonstration that a carefully constructed, permissive-first mixture can achieve competitive downstream performance without relying on legally ambiguous data and release/documentation of the dataset artifacts.

**Strengths:**

(1) Thorough, transparent, and literature-grounded curation pipeline. \
The paper provides a clear and well-justified account of how the dataset was constructed, with each design choice motivated by established best practices in large-scale data curation and LLM training. The authors’ approach is thoroughly grounded in prior literature and consistent with contemporary standards, reflecting a careful, research-driven methodology rather than ad-hoc engineering.

(2) Careful empirical validation, principled source selection, and clear legal rationale. \
The authors conduct rigorous and systematically controlled experiments, maintaining fixed architectures and hyperparameters to isolate the effects of data quality and composition. More broadly, the experimental design reflects an academic level of care in controlling confounding factors and articulating how scaling interacts with dataset construction. The dataset sources are thoughtfully selected, with explicit reasoning about what constitutes “permissive” or “low-risk” content. The tiered legal framework is well documented and well-reasoned, reflecting a careful effort to maximize the research utility of the data while appropriately managing potential legal and compliance risks.

(3) Careful decontamination analysis that convincingly addresses leakage concerns. \
Appendix D.3 and Table 9 show downstream test set exact-match filtering, overlap statistics, and evaluation with highly contaminated shards/documents removed. This analysis alleviates the most natural skepticism, that high performance stems from test overlap rather than data quality.

(4) High-quality writing and presentation. \
The paper is readable, well-structured, and supported by comprehensive experiments and reproducibility documentation.

**Weaknesses:**

(1) Synthetic-data provenance and generation details remain somewhat opaque. \
If I understand correctly, the authors do not themselves generate new synthetic data but instead aggregate existing permissively licensed datasets that may include synthetically generated data. However, some of these sources (e.g., Nemotron-CC, MagaCorpus) were produced by rephrasing or re-expressing web text from non-permissive sources. It would help to clarify exactly how these synthetic subsets were created—whether they consist primarily of paraphrases of non-permissive text, newly generated reasoning or math/code problems, or continuations of conversational data. More broadly, is there existing evidence (either from the authors or prior literature) that such rephrasing or rewriting pipelines reliably avoid leakage of protected content? Clarifying these points would strengthen confidence in the “permissive-only” guarantee, though this is more a request for detail than a critique of the current results.

(2) Organization and emphasis of test data contamination results. \
The ablation analysis in Section 3.4 is somewhat coarse, focusing primarily on Full, without Web, without Instructions. Given the breadth of the results, this section could be condensed or moved to the Appendix, with more space in the main body dedicated to the data contamination analysis (Appendix D.3 and Table 9). The possibility of test set leakage is one of the most plausible alternative explanations for the paper’s strong capability results, and the contamination analysis directly and convincingly addresses that concern. Bringing more of this evidence into the main text would highlight how the authors empirically address one of the strongest counter-hypotheses, thereby strengthening the credibility of the central claims.

(3) Contamination detection focuses on exact n-gram matches. \
The 13-gram decontamination method used is standard but may overlook paraphrased or semantically similar examples. It would be helpful for the authors to comment on whether they explored or plan to explore approximate or embedding-based methods (e.g., LSH, SimHash, ANN) for identifying near-duplicates or semantic overlaps. Even a brief discussion of these trade-offs would further strengthen the already careful analysis in Appendix D.3.

(4) Limited granularity in reasoning/instruction ablations. \
If the dataset is already organized into domain-aware shards, there may be value in providing a finer-grained ablation within the reasoning and instruction components themselves. As currently presented, the without Instructions ablation yields an intuitive result, performance on MMLU and related reasoning benchmarks drops sharply, whereas without Web has a much smaller effect. A more granular analysis of which shards within the reasoning/instruction data (e.g., math, coding, general instruction, synthetic Q&A) contribute most to downstream performance would be more informative and would better illuminate what kinds of reasoning data are most valuable.

**Questions:**

1. Could the authors clarify how the synthetic subsets were actually produced? From the description, it seems they are drawn from existing datasets with synthetically generated data rather than newly generated data. Were these primarily rephrasings of non-permissive text, continuations of such text in a chat or QA style, or newly constructed reasoning and math examples designed to resemble those domains? Understanding this distinction would help clarify how these datasets were created and why they are considered permissive.

2. Am I correct in understanding that the dataset shards are formed based on source-domain groupings or other dataset-level partitions (e.g., those listed in Table 11)? While I understand that additional ablations may be resource-intensive, it struck me that it would be extremely valuable for the research community if the released dataset included clear documentation of how these shards are defined and organized—along with their relative sizes, license tiers, and token counts. Making this shard-level metadata available would enhance transparency and enable future research/ablations on data mixtures.

3. The downstream results show that the chosen reasoning, math, and coding datasets were remarkably effective in driving strong benchmark performance, even though the mixture appears to rely on a relatively focused set of high-quality permissive sources. Could the authors elaborate on how these particular datasets were selected? Were they chosen to be representative of the domains emphasized in downstream benchmarks, or based on prior evidence or community popularity in other pretraining mixtures?

Small line-level edits
- Line 110: "It also contain synthetic data" -> "It also contains synthetic data"
- Line 122: I think the Saxton in-line citation is misformatted.
- Line 142: "as well domain-specific datasets" -> "as well as domain-specific datasets"

---

> ### Author Response · Authors · 2025-11-21
>
> We thank the reviewer for their detailed assessment and for specifically highlighting our 'thorough, transparent' curation pipeline and our 'careful decontamination analysis' that 'convincingly addresses leakage concerns.' We address the specific concerns below.
>
> ### Organization and emphasis of test data contamination results
> This is a great point and we fully agree.
>
> **Action**: we will update the main text to highlight the contamination analysis.
>
> ### Approximate or Embedding-Based Decontamination Methods
>
> We thank the reviewer for this thoughtful suggestion. Our current decontamination pipeline follows the standard 13‑gram exact‑match procedure used in recent data‑centric work such as Phi, primarily for two reasons: (i) high precision: we want to be certain that any flagged overlap corresponds to near‑verbatim reuse of a test sequence; and (ii) comparability and scalability: 13‑gram hashing scales easily to hundreds of billions of tokens and aligns with prior evaluations. We agree, however, that this method does not capture all paraphrased or semantically similar examples.
>
> We did not deploy approximate or embedding‑based methods (LSH, SimHash, ANN) for test decontamination in MixtureVitae, mainly due to the precision-recall trade‑off: at realistic thresholds these methods tend to produce many false positives, especially for common factual snippets, standard algorithm templates, or textbook‑like phrasing. At our scale, aggressively removing all “semantically similar” content risks distorting the training distribution by deleting a large amount of legitimate high‑value data.
>
> **Action**: We will update the appendix to include this discussion.
>
> ### Dataset Shard Formation
>
> To clarify: the high-level groupings presented in the paper (e.g., "Code & Tech," "Reasoning”) and the shard breakdown in Table 11 are primarily organizational abstractions intended to help the reader visually parse the dataset's composition .
>
> In practice, the actual training data construction follows a domain-aware mixing strategy inspired by Phi-4: Rather than training on rigid "shards" sequentially, we cluster documents by their base URL or provenance to preserve domain coherence per sample. Our curation strategy mirrors the "textbook quality" approach of Phi-4, where the priority is not domain grouping, but the specific density of reasoning and factual tokens.
>
> **Action**: We will add a clarifying note to Section 2.1 and Appendix A to distinguish between the visual taxonomy used for reporting and the granular mixing strategy used for training.
>
> ### Granular Reasoning Ablations
>
> We agree that further isolating the contribution of specific reasoning subsets (e.g., Math vs. Code vs. QA), in addition to ablations on larger Web and Instruction subsets performed in Fig. 4, would be valuable. However, given the computational cost of pre-training at the 100B+ scale, running separate ablations for every sub-shard was infeasible for this study, while our results already provide strong evidence that the Reasoning & Instructions subset is responsible for our strong performance on math and code (Fig 4b) To empower the community to further investigate more specific smaller subsets, we release all shard metadata: token counts, license tiers, and source categories, enabling future fine-grained ablation studies.
>
> ### Dataset Selection
> We selected datasets based on "permissive-first" criteria followed by quality and diversity objectives. Our selection process involved two distinct layers of criteria:
> * **Legal & Licensing**: The primary filter was legal. A dataset was only considered if it was under a clear permissive license (e.g., CC-BY) or in the public domain. For synthetic data, we furthermore prioritized a permissive provenance of seed corpora and generator models. The large majority of our synthetic sources satisfy this and are classified as Tier‑1. A small number of widely used community reasoning datasets do not provide full seed data provenance. We include these under their permissive dataset licenses but classify them as Tier‑2 (low‑but‑non‑zero residual risk).
> * **Size, Quality & Capability**: Among the sources that passed the legal filter, we prioritized datasets based on:
>    - **Prior Evidence of Performance**: We extensively leveraged datasets that have become community standards in other high-performance mixtures (e.g., peS2o).
>    - **Reasoning & Diversity**: We specifically selected data to boost reasoning and domain coverage where strictly permissive web scrapes are traditionally weak. This motivated the inclusion of reasoning and instruction data, which is known to boost performance on tasks like GSM8K and MMLU.
>
> In summary, selection was dictated by the intersection of permissive licensing and the need for high-density reasoning data to ensure performance.
>
> **Action**: We will incorporate a description of this selection process into the Methodology section.

---

> > ### Author Response · Authors · 2025-11-21
> >
> > ### Synthetic Dataset Provenance
> >
> > We appreciate the reviewer’s request for more detail on how the synthetic subsets were produced and how they fit with our permissive‑first philosophy. To clarify, MixtureVitae does not treat all synthetic data as a single homogeneous block. Instead, we distinguish three categories:
> >
> > **(A) Web‑rewrite permissive corpora with domain allowlisting (Tier-1)**
> >
> > For datasets like Nemotron-CC, we only retain samples where the underlying source URL matches our permissive allowlist (.gov, CC-BY), as detailed in Section 2.2.1. This ensures that rephrasing is not used to launder non-permissive content or circumvent opt-outs. This means that even when the synthetic pipeline is conceptually a “rephrasing” of web content, the inputs are already permissive or government works, so the rewrites do not serve as a way to circumvent opt‑outs or scrape rights‑reserved outlets.
> >
> > **(B) Reasoning/math/code datasets built from permissive sources or fully synthetic seeds (Tier-1)**
> >
> > Datasets generated by permissive models from explicitly permissive seeds (e.g., Dart-Math). These contain newly constructed reasoning traces, explanations, and QA‑style problems. They form the bulk of our “Reasoning/Instruction” block in Figure 1. We classify them as Tier‑1 when both the dataset license, the seed text and the generator model are under permissive licenses.
> >
> > **(C) Community reasoning corpora with partially opaque seeds (Tier‑2)**
> >
> > Finally, we included a small number of community datasets where the subset is permissively licensed, but the seed data or the generated model may have a higher risk provenance than (A) or (B). Combined, these add up to under 3% of MixtureVitae.
> >
> > **On whether rephrasing “eliminates” copyright concerns.**
> >
> > We agree that rephrasing alone is not a formal guarantee against leakage of protected content. Our risk mitigation does not rely solely on style transfer: for web‑rewrite corpora we additionally require that the seed domains themselves be permissive or governmental, and for synthetic reasoning datasets we either (i) verify that the underlying sources are permissively licensed, or (ii) if that is not possible or there is ambiguity, classify them explicitly as Tier‑2. This is why throughout the paper we describe MixtureVitae as “permissive‑first, risk‑mitigated” rather than “strictly permissive‑only”.
> >
> > We hope that this more detailed breakdown addresses the reviewer’s request: most of our synthetic data is either (i) rewrites and QA over already‑permissive/governmental corpora or (ii) newly constructed reasoning/math/code traces over permissive datasets, and a small set of partially opaque community datasets is carried as Tier‑2.
> >
> > **Action**: We will update Section 2 and the Appendix to explicitly define these three provenance categories and document the specific datasets belonging to Tier-2/3 to ensure transparency regarding their provenance. This explicit categorization should allow users to easily exclude these tiers if needed.
> >
> > Once again, we thank the reviewer for the the thorough review and helpful feedback and we hope our detailed responses have addressed all outstanding concerns.

---

### Author Response · Authors · 2025-11-20
**General Comment for all Reviewers**

We thank the reviewers for their constructive feedback and for recognizing the significance of a permissive‑first corpus. We will revise the manuscript to incorporate the reviewers’ suggestions and clarify points that were previously unclear. We are encouraged by the consensus on our core strengths:

* **Timely Contribution**: Described as "timely and commendable" (otHz) and addressing a "critical problem" (1Bso) by demonstrating high-performance training with "legally safer" (otHz) sources.
* **Transparent Curation**: The pipeline was noted as "thorough, transparent, and literature-grounded" (NdNQ) and "well documented" (1Bso), with a decontamination analysis that "convincingly addresses leakage concerns" (NdNQ).
* **Strong Validation**: Reviewers praised the "rigorous experiments" (otHz), noting MixtureVitae "outperforms all other permissive alternatives" (otHz) and "approaches parity" (NdNQ) with non-permissive baselines.

Across the reviews, we identified a few shared themes, which we address below: **Dataset Scalability, Curation Strategy & Licensing, and Evaluation Integrity: Contamination and Memorization**.

### Dataset Size and Scalability

Regarding the observed scale difference (211B vs 10T+ in frontier runs), our primary goal was a proof-of-concept for data efficiency and strong performance within a permissive‑first, risk‑mitigated licensing framework. We see several avenues to scale this approach:
* **Clarification on Baselines**: Common Pile is an 8TB pool; the derived training set (Comma-0.1) is ~600B unique tokens.
* **Subset Upsampling**: Unlike common recipes (e.g., Llama3) that upsample high-quality subsets 4-10x, MixtureVitae does not assign special upsampling factors to individual shards. Upsampling our highest-value subsets would immediately scale their contribution.
* **Multilingual Expansion**: Expanding to multilingual data, via either sourcing additional datasets or translation, represents an order-of-magnitude opportunity for scaling.
* **Synthetic Expansion**: Our synthetic subsets could be expanded by generating more samples using permissively licensed models.
* **Web Data Rephrasing**: Rephrasing of web data, subject to the same allowlist andlegal‑risk framework we use in MixtureVitae, can help expand the corpus.

### Curation Strategy & Licensing

Reviewers (OtHZ, 1Bso) discussed our data curation and licensing methodology. We wish to clarify our "Permissive-First, Risk-Mitigated" design philosophy, which distinguishes MixtureVitae from large-scale crawls like Common Corpus or Apertus.
* Pipelines such as Apertus operate on broad web scrapes and then apply negative filtering (e.g., discarding data based on current `robots.txt` rules).
* **Our Approach**: MixtureVitae uses a positive inclusion (allowlist) strategy. We positively selected safe sources derived from Common Crawl. In regard to opt-out, we leverage Common Crawl’s inherent adherence to `robots.txt` at the time of capture.
* **Benefit**: By filtering the web data component of MixtureVitae for permissive and `.gov` (or similar) domains, and leveraging Common Crawl’s inherent adherence to opt-out protocols, we believe we comply with text data mining opt-out regimes and guard against the ingestion of copyrighted content (like the New York Times) from the start. This proactive approach substantially reduces reliance on retroactive decontamination and, together with our 13-gram decontamination checks, establishes a cleaner legal profile for users.
* **Tiered Legal‑Risk Framework and Synthetic Data**: As described in Section 2.1.4, we additionally organize all sources, including synthetic instruction/reasoning data, into tiers with distinct risk profiles. Most of MixtureVitae’s data has a clearly permissive provenance and falls into our lowest‑risk tier (Fig. 2(b)).

### Evaluation Integrity: Contamination and Memorization
Reviewers emphasized the importance of ruling out test leakage and, more broadly, the risk of memorization on restricted sources (e.g., NYT). We performed a 13‑gram decontamination sweep over all benchmarks: document‑level overlap is negligible for most tasks, and performance is stable when we (i) re-evaluate on decontaminated test sets (Table 9) and (ii) retrain after removing shards that account for most of the contamination signal (Fig. 9).

Because >90% of MixtureVitae consists of code, math, synthetic reasoning, and curated scientific and civic sources, and the remaining web component is built via positive inclusion and license‑keyword filtering, we expect the presence of paywalled or opted‑out publishers to be very low. As a result, NYT‑style memorization tests are less diagnostic for this dataset than for broad web scrapes, though we agree that such tests are valuable future work.

We are thankful for the reviewers for their time and constructive engagement. We will use the feedback to strengthen our manuscript and we hope that our responses have addressed all outstanding concerns.

---

### Author Response · Authors · 2025-11-26
**Summary of Changes to Manuscript**

We thank all reviewers for their detailed and constructive feedback. Below we summarize the concrete changes made in response to each review.

### Reviewer NdNQ

**Synthetic-data provenance.**
Added a detailed synthetic provenance table (Table 13, App. J) specifying dataset license, generator model, seed provenance, and assigned tier.

**Contamination analysis.**
Introduced Section 3.5 “Contamination and leakage”, which provides a short summary of the decontamination protocol that is detailed in the Appendix.

**13-gram vs approximate decontamination.**
Added Appendix D.5, explaining why we use 13-gram exact matching (precision, scalability, comparability with prior work) and discussing the trade-offs and false-positive risks of approximate / embedding-based decontamination at our scale.

### Reviewer 1Bso

**Dataset size and scalability.**
Added Appendix K outlining concrete scaling paths (subset upsampling, multilingual expansion, synthetic growth, web rephrasing) toward multi-trillion-token regimes.

**Comparison with Apertus.**
Expanded Section 4.3 to explicitly compare MixtureVitae and Apertus.

### Reviewer otHz

**License checking and web compliance.**
Clarified our positive-inclusion, allowlist-based pseudo-crawl in Section 2.3 and Appendix I.

**Fairness of reasoning-heavy pretraining vs web-only baselines.**
Updated Section 3.3 to clearly describe MixtureVitae as a reasoning/instruction-heavy pretraining mixture and to state that our goal is to isolate the effect of changing only the dataset under fixed architectures and budgets.

Added an explicit comparison to SmolLM2-1.7B-Instruct in Table 1 and text, noting that our 1.7B base model (300B tokens) matches or exceeds it on GSM8K, HumanEval, and MBPP despite SmolLM2-Instruct using  roughly 11T tokens and post-training.

**Dataset scale vs Common Pile.**
The scaling discussion in Appendix K explains how the MixtureVitae recipe can be extended beyond the current open-sci-ref setting.

We hope this summary helps explain how the manuscript changes respond to the concerns that were raised, and we appreciate the feedback that enabled us to improve our paper.

---

### Author Response · Authors · 2025-11-28
**Update: Dataset Size Accounting**

We would like to provide a brief clarification regarding the dataset size reported in the original submission.

During a post-review audit, we discovered a counting error in our corpus-statistics script that uniformly underreported token counts across all components. The dataset used in all experiments is ~422B tokens rather than the ~211B originally stated. Because the error affected all components proportionally, **the mixture structure and per-subset token counts relative to one another remain exactly the same**.

This correction strengthens the paper’s position in regard to dataset scale:
_At ~422B tokens, MixtureVitae is now comparable in size to Comma-0.1 (which contains ~600B unique tokens), helping address concerns about whether the dataset was too small relative to other permissive baselines._

Importantly:
* **The training protocol is unchanged**: all models were trained for 50B and 300B tokens under the open-sci-ref setup, and the counting error was confined exclusively to the corpus-statistics script.
* **All experiments were done on fixed token budgets** and therefore all results, comparisons and conclusions remain the same.

We have updated the descriptive corpus statistics (e.g., Figure 1, Table 2, and related text) to reflect the correct dataset size. No experiments were rerun, and all performance claims remain valid.

We appreciate the opportunity to clarify this point and remain committed to accurate and transparent reporting.

---

### Note · Program_Chairs · 2025-12-08
**Submission Desk Rejected by Program Chairs**

Desk rejected because of hallucinated citations:
Dan Hendrycks, Collin Burns, Steven Basart, Andy Critch, Jerry Li, Dawn Ippolito, Aina Lapedriza,
Florian Tramer, Rylan Macfarlane, Eric Jiang, et al. Measuring massive multitask language un-
derstanding. In Proceedings of the International Conference on Learning Representations (ICLR),
2021.